# Neuropilin 1 regulates bone marrow vascular regeneration and hematopoietic reconstitution

Christina M. Termini [1,2,3], Amara Pang[2], Tiancheng Fang[1,4], Martina Roos[1,5,6], Vivian Y. Chang[6,7], Yurun Zhang[1,8], Nicollette J. Setiawan[2], Lia Signaevskaia[1], Michelle Li[1], Mindy M. Kim[1], Orel Tabibi[1], Paulina K. Lin[1], Joshua P. Sasine [1,2], Avradip Chatterjee[9], Ramachandran Murali[9], Heather A. Himburg[10] & John P. Chute[2,11,12 ✉]

Ionizing radiation and chemotherapy deplete hematopoietic stem cells and damage the vascular niche wherein hematopoietic stem cells reside. Hematopoietic stem cell regeneration requires signaling from an intact bone marrow (BM) vascular niche, but the mechanisms that control BM vascular niche regeneration are poorly understood. We report that BM vascular endothelial cells secrete semaphorin 3 A (SEMA3A) in response to myeloablation and SEMA3A induces p53 – mediated apoptosis in BM endothelial cells via signaling through its receptor, Neuropilin 1 (NRP1), and activation of cyclin dependent kinase 5. Endothelial cell – specific deletion of *Nrp1* or *Sema3a* or administration of anti-NRP1 antibody suppresses BM endothelial cell apoptosis, accelerates BM vascular regeneration and concordantly drives hematopoietic reconstitution in irradiated mice. In response to NRP1 inhibition, BM endothelial cells increase expression and secretion of the Wnt signal amplifying protein, R spondin 2. Systemic administration of anti - R spondin 2 blocks HSC regeneration and hematopoietic reconstitution which otherwise occurrs in response to NRP1 inhibition. SEMA3A – NRP1 signaling promotes BM vascular regression following myelosuppression and therapeutic blockade of SEMA3A – NRP1 signaling in BM endothelial cells accelerates vascular and hematopoietic regeneration in vivo.

[1] Division of Hematology/Oncology, Department of Medicine, University of California, Los Angeles, CA, USA. [2] Division of Hematology & Cellular Therapy, Cedars Sinai Medical Center, Los Angeles, CA, USA. [3] Department of Orthopedic Surgery, UCLA, Los Angeles, CA, USA. [4] Department of Molecular and Medical Pharmacology, UCLA, Los Angeles, CA, USA. [5] Eli and Edythe Broad Stem Cell Research Center, UCLA, Los Angeles, CA, USA. [6] Jonsson Comprehensive Cancer Center, UCLA, Los Angeles, CA, USA. [7] Pediatric Hematology/Oncology, UCLA, Los Angeles, CA, USA. [8] Molecular Biology Institute, UCLA, Los Angeles, CA, USA. [9] Department of Biomedical Sciences, Research Division of Immunology, Los Angeles, USA. [10] Department of Radiation Oncology, Medical College of Wisconsin, Milwaukee, WI, USA. [11] Regenerative Medicine Institute, Cedars Sinai Medical Center, Los Angeles, CA, USA. [12] Samuel Oschin Cancer Center, Cedars Sinai Medical Center, Los Angeles, CA, USA. ✉email: john.chute@cshs.org

HSC maintenance requires complex interactions between HSCs and their BM microenvironment, or niche, involving signals from perivascular stromal cells, vascular endothelial cells (ECs), sympathetic nervous system, macrophages, and megakaryocytes[1–11]. HSC regeneration is a process that is necessary for blood system recovery following medically relevant insults such as chemotherapy, irradiation, infection, and chronic inflammation[12–18]. The mechanisms governing HSC regeneration are not well understood, but it has been shown that BM ECs are essential for HSC regeneration to occur following myelosuppression[19–22]. We have also shown that BM ECs secrete growth factors, including pleiotrophin and epidermal growth factor, that are essential for HSC regeneration following total body irradiation (TBI)[12,23–25]. Chemotherapy and irradiation also severely damage the BM vasculature[13,19,24,26] and the mechanisms that govern BM vascular regeneration following myeloablation are incompletely understood[26,27]. We recently discovered that BM ECs increase expression of genes that encode semaphorin (SEMA) proteins, including Class III secreted SEMA proteins, within 96 h following TBI[28]. SEMAs are extracellular signaling proteins that have a common cysteine-rich extracellular domain[29] and were first described as axon guidance molecules[29,30]. During development, SEMA proteins direct the migration and segregation of neural crest cells[31], control the migration of cortical neurons and cerebellar neurons and establish the boundary between the central and peripheral nervous systems[29,31–34]. In the adult, SEMA proteins regulate neuronal proliferation and migration and provide repulsive and attractive forces for axons, mediated via binding to their receptors, plexins and neuropilins (NRPs)[29,35–38].

During development, Class 3 SEMA proteins have been shown to regulate the formation of the dorsal aorta and the embryonic heart, whereas SEMA4D promotes angiogenesis in implanted chick embryos[29,39–45]. Endothelial tip cells that guide blood vessel patterning and angiogenesis also express NRP and plexin receptors for SEMA proteins[29,42,46]. Class 3 SEMAs have additionally been shown to suppress angiogenesis in a chick chorioallantoic membrane assay[47] and to regulate vascular permeability via in vitro assays[48,49]. Neuropilin-1 has been shown to be expressed by BM stromal cell lines, but the role of semaphorin signaling in regulating BM vasculogenesis or normal hematopoiesis is not known[50].

Here, we show that myeloablation causes BM ECs to upregulate expression of SEMA3A, and its receptor, NRP1, and SEMA3A-NRP1 signaling promotes BM EC apoptosis and delayed BM vascular regeneration in myeloablated mice. Genetic or pharmacologic inhibition of SEMA3A-NRP1 signaling increases BM EC survival and accelerates BM vascular regeneration in myeloablated mice and concordantly drives HSC regeneration and early hematopoietic reconstitution.

## Results

**BM ECs increase expression of SEMA3A and NRP1 following TBI**. We discovered that the expression of several genes encoding SEMA proteins, including Sema3a, Sema3f, Sema6a, and Sema7a, increased significantly in VE-cadherin+ (VE-cad+) BM ECs by array analysis within 24 h after 500 cGy TBI[28]. RNAseq analysis of BM CD45-VE-cad+ BM ECs demonstrated increased expression of Class 3 semaphorin genes, Sema3a, Sema3b, and Sema3d at 72 h following 500 cGy TBI and quantitative RT-PCR analysis confirmed that Sema3a expression increased in CD45-VE-cad+ BM ECs following 500 cGy TBI, along with genes encoding the Class 3 semaphorin receptors, Neuropilin 1 (Nrp1) and 2 (Fig. 1a, b)[51]. At baseline, Sema3a expression was highest in CD31+Sca-1− sinusoidal BM ECs (sBMECs) and CD31+Sca-1+ arteriolar BM ECs (aBMECs), and detected at very low levels in whole BM

cells, lin− cells and ckit+sca-1+lin− (KSL) hematopoietic stem/progenitor cells (HSPCs) (Fig. 1c and Supplementary Fig. 1a). Sema3a expression was not detected in Leptin receptor (LepR+) stromal cells (Fig. 1c). Following 500 cGy TBI, Sema3a gene expression increased in aBMECs and remained elevated in sBMECs (Fig. 1d). Nrp1 expression was also more than 100-fold higher at baseline in sBMECs and aBMECs compared to BM hematopoietic cells and LepR+ stromal cells (Fig. 1e). Following 500 cGy TBI, we detected substantially increased levels of SEMA3A protein in the BM of mice over time compared to non-irradiated controls (Fig. 1f). In vitro irradiation of BM ECs with 800 cGy also increased SEMA3A protein levels in culture after 24 h (Supplementary Fig. 1b). Microscopic analysis of femurs from non-irradiated C57BL/6J mice confirmed co-localization of SEMA3A expression with BM ECs that co-expressed VE-cad+ and CD31+ (Fig. 1g). At 72 h following 500 cGy TBI, SEMA3A expression increased, localized to VE-cad+CD31+ BM ECs. NRP1 expression also co-localized with VE-cad+CD31+ BM ECs in non-irradiated mice and increased following 500 cGy TBI (Fig. 1h). Flow cytometric analysis confirmed that NRP1 protein expression increased on CD45−CD31+ BM ECs and on sBMECs at 24 h following 500 cGy TBI (Fig. 1i, j), whereas NRP1 expression remained high on aBMECs before and after TBI (Fig. 1j)[8]. Mean fluorescence intensity (MFI) of NRP1 surface expression increased on CD31+ BM ECs, aBMECs and sBMECs at 24 h following 500 cGy TBI (Fig. 1k). These results suggest that BM ECs differentially express SEMA3A and its receptor, NRP1, compared to other BM cells, and increase secretion of SEMA3A and surface expression of NRP1 in response to TBI.

**Inhibition of NRP1 accelerates BM vascular regeneration**. Since TBI induced the secretion of SEMA3A and expression of its receptor, NRP1, by BM ECs, we tested the effect of inhibition of SEMA3A-NRP1 signaling on the BM vascular response to TBI. For these studies, we treated mice with goat anti-mouse NRP1 antibody or isotype IgG control antibody. Baseline microscopic imaging of the BM vasculature in non-irradiated adult C57BL/6J mice is shown in Fig. 2a. TBI with 500 cGy caused dilatation and disruption of the sinusoidal BM vasculature within 72 h and loss of BM vascular density at day +7 in isotype-treated control mice (Fig. 2b). Irradiated mice treated with anti-NRP1 antibody (10 μg) intraperitoneally (IP) every other day from day +1 to day +10 also demonstrated BM vascular dilatation at day +3 similar to irradiated control mice; however, by day +7, anti-NRP1 treated mice displayed nearly complete reestablishment of the BM sinusoidal architecture (Fig. 2b)[52]. Irradiated, control mice displayed increased BM vascular area, which is an indicator of BM vascular damage[26], at day +3 and day +7 following 500 cGy TBI; irradiated mice treated with anti-NRP1 demonstrated increased BM vascular area at day +3, but normalization of BM vascular area by day +7 (Fig. 2c). Irradiated, anti-NRP1 treated mice also demonstrated increased BM cell counts and increased percentages and numbers of VEcad+ BM ECs at day +7 following TBI compared to irradiated controls (Fig. 2d, e and Supplementary Fig. 1c)[24]. Irradiated control mice displayed more than 3-fold increased percentages of apoptotic VE-cad+ BM ECs, based on activated Caspase 3/7 staining, compared to non-irradiated mice (Fig. 2f). Irradiated mice treated with anti-NRP1 demonstrated significantly decreased percentages of apoptotic BM ECs compared to irradiated controls. Of note, we observed no significant differences in the cell cycle status of BM ECs between irradiated, control mice and irradiated, anti-NRP1 treated mice at day +5 following 500 cGy TBI (Supplementary Fig. 1d, e).

At +24 h following 500 cGy TBI, control mice also demonstrated increased BM extravasation of Evans Blue Dye (EBD)

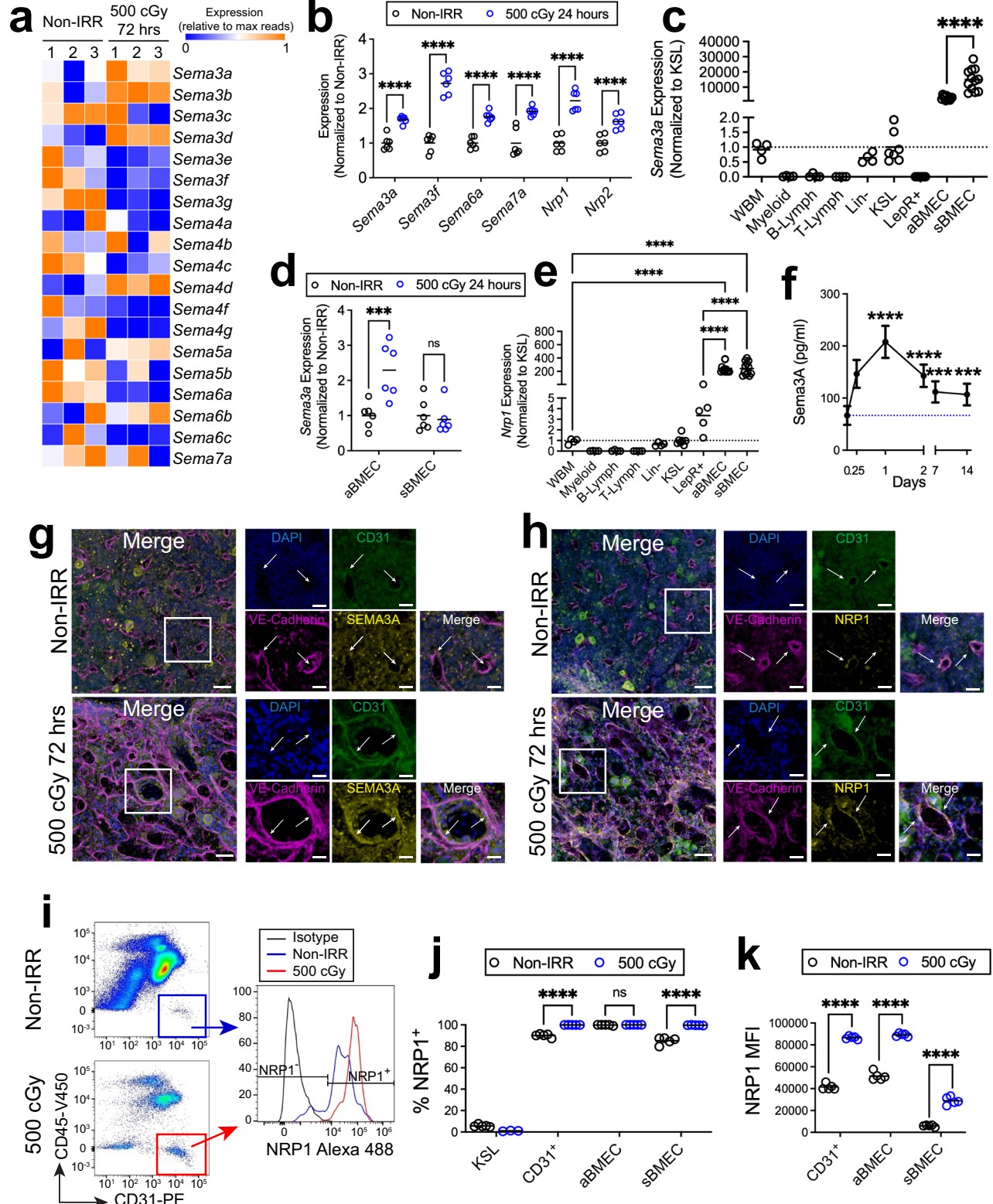

compared to non-irradiated mice, an indication of decreased vascular integrity (Fig. 2g)[27]. In contrast, irradiated mice treated with anti-NRP1 demonstrated no increase in EBD extravasation compared to non-irradiated mice (Fig. 2g). In order to assess whether inhibition of SEMA3A-NRP1 signaling affected BM vascular perfusion following TBI, we intravenously injected *C57BL/6J* mice with 100 μg tomato-lectin on day +7 following 500 cGy TBI. We detected no differences in BM vascular

perfusion between irradiated control mice and irradiated, anti-NRP1 treated mice (Supplementary Fig. 1f, g).

In complementary studies, we evaluated the effects of systemic administration of recombinant SEMA3A on BM vascular recovery in mice following TBI. Here, we administered 2 μg of recombinant human full length SEMA3A Fc (huSEMA3A, Lys26-Val771) which has >95% sequence homology with murine SEMA3A, contains the highest affinity binding sites for NRP1,

**Fig. 1 BM ECs upregulate expression of SEMA3A and its receptor, NRP1, following irradiation. a** Heatmap of semaphorin (Sema) gene expression in irradiated and non-irradiated (non-IRR) BM ECs isolated before and after irradiation ($n = 3$/condition). **b** qRT-PCR analysis of Sema and Nrp expression in VE-cad$^+$ BM ECs from Non-IRR and 500 cGy-irradiated mice ($n = 6$/condition from $n = 10,000$ cells/replicate; ****$p < 0.0001$). Data normalized to non-IRR sample. Horizontal bars represent means. **c** qRT-PCR of Sema3a expression in BM hematopoietic cell populations, LepR$^+$ stromal cells and CD31$^+$Sca-1$^-$ sinusoidal BM ECs (sBMECs) and CD31$^+$Sca-1$^+$ arterial BM ECs (aBMECs; $n = 4$–12 replicates/condition; ****$p < 0.0001$). Data normalized to KSL cells. **d** qRT-PCR of Sema3a expression in aBMECs and sBMECs before and +24 h following 500 cGy ($n = 6$/group; aBMEC: ***$p = 0.0008$, sBMEC: n.s. $p = 0.72$). Data normalized to non-IRR. **e** qRT-PCR of Nrp1 expression in BM hematopoietic cell populations, LepR$^+$ stromal cells and ECs ($n = 4$–12 replicates/group; ****$p < 0.0001$). Data normalized to KSL cells. **f** SEMA3A concentration in the BM of mice following 500 cGy quantified by ELISA ($n = 8$ mice; 6 h, d1, d2: ****$p < 0.0001$, d7: ***$p = 0.0001$, d14: ***$p = 0.0007$). Dotted line represents BM SEMA3A concentration in non-IRR mice. **g** Confocal images of femurs from C57BL/6J mice before and +72 h following 500 cGy. Expression of SEMA3A (yellow) in VE-Cad$^+$ BM ECs (magenta) and CD31$^+$ BM ECS (green) is shown (white arrows). Nuclei labeled with DAPI (blue). Representative images from $n = 3$ experiments, Scale bar 50 μm; magnified scale bar, 20 μm; max z-projections shown. **h** Expression of NRP1 (yellow) in VE-Cad$^+$ BM ECs (magenta) and CD31$^+$ BM ECS (green) is shown (white arrows) in same conditions. Representative images from $n = 3$ experiments. **i** Representative gating, histogram, and **j** quantification of NRP1 expression on Non-IRR and 500 cGy-irradiated BM ECs ($n = 5$/group, ****$p < 0.0001$, aBMEC: $p = 0.8256$). **k** NRP1 MFI within BM EC sub-populations before and +24 h post-500 cGy ($n = 5$/group, ****$p < 0.0001$). Data assessed by Holm-Sidak's multiple comparison two-sided $t$-test after two-way ANOVA (**b, d, f, j, k**) and the Holm-Sidak's multiple comparison two-sided $t$-test after one-way ANOVA (**c, e**). Data presented as mean values +/− SEM. **$p < 0.01$, ***$p < 0.001$, ****$p < 0.0001$. Source data are provided as a Source Data file. .

and displays high affinity binding to murine NRP1 (Supplementary Fig. 2a–d), intraperitoneally (IP), every other day from day +1 to day +10 following 500 cGy TBI in C57BL/6 J mice. We also administered recombinant mouse SEMA3A Fc (muSEMA3A, Asn21-Lys747) to irradiated mice. Irradiated mice treated with huSEMA3A or muSEMA3A displayed no additional disruption of the BM vasculature and change in BM vascular area beyond that observed in irradiated, vehicle-treated control mice (Supplementary Fig. 2e). Therefore, systemic administration of SEMA3A with high binding affinity to NRP1 did not worsen TBI-induced damage to the BM vasculature in mice.

**NRP1 inhibition accelerates hematopoietic reconstitution.** Since anti-NRP1 treatment promoted early restoration of the BM vasculature in irradiated mice, we tested whether anti-NRP1 treatment could also facilitate hematopoietic regeneration. C57BL/6J mice irradiated with 500 cGy TBI and treated with IgG demonstrated peripheral blood (PB) leukopenia, neutropenia and lymphocytopenia at day +10 (Fig. 3a). In contrast, irradiated, anti-NRP1-treated mice demonstrated early recovery of PB white blood cells (WBCs), neutrophils, and lymphocytes over time compared to irradiated IgG-treated mice (Fig. 3a). Irradiated, anti-NRP1-treated mice also demonstrated accelerated recovery of BM cell counts, percentages and numbers of BM c-kit$^+$sca-1$^+$lin$^-$ (KSL) HSPCs[53], ckit$^+$sca-1$^-$lineage$^-$ myeloid progenitor cells, and BM colony forming cells (CFCs) at day +10 compared to irradiated controls (Fig. 3b, c and Supplementary Fig. 3a, b). Systemic administration of huSEMA3A to irradiated mice did not significantly affect the recovery of PB complete blood counts, BM cell counts, or BM KSL cells compared to irradiated control mice (Supplementary Fig. 4a–d). Of note, we also irradiated BM KSL cells in vitro with 300 cGy and treated for 7 days with 10 μg/ml anti-NRP1 or IgG and observed no direct effects of anti-NRP1 antibody treatment on hematopoietic cell counts or KSL percentages after irradiation (Supplementary Fig. 4e). These results suggested that anti-NRP1 caused no direct effects on HSPCs.

In order to determine the effects of anti-NRP1 treatment on the regeneration of BM HSCs in irradiated mice, we performed competitive repopulation assays using donor CD45.2$^+$ BM cells collected at day +10 from irradiated C57BL/6J mice, treated with IgG or anti-NRP1, and transplanted $5 \times 10^5$ donor BM cells, along with $2 \times 10^5$ competitor CD45.1$^+$ BM cells, into recipient CD45.1$^+$ B6.SJL mice (Fig. 3d). Mice transplanted with BM cells from irradiated, anti-NRP1-treated mice demonstrated significantly increased donor CD45.2$^+$ cell engraftment at 20 weeks of post-transplant compared to mice transplanted with the identical dose of BM cells from irradiated, IgG-treated mice (Fig. 3d).

Donor myeloid cell, B cell and T cell chimerism was also significantly increased, as was donor cell engraftment within the BM KSL HSPC population, in recipients of BM cells from irradiated, anti-NRP1 treated mice compared to recipients of BM cells from irradiated, control mice (Fig. 3d, e). Secondary competitive BM transplantation assays using $3 \times 10^6$ BM cells collected at 20 weeks from primary recipient mice and $2 \times 10^5$ competitor BM cells did not demonstrate a significant increase in donor cell engraftment in recipient mice in the anti-NRP1 treatment group compared to controls (Supplementary Fig. 4f).

In order to determine if systemic anti-NRP1 treatment could provide radioprotection to lethally irradiated mice, we irradiated C57BL/6J mice with 800 cGy TBI and treated with or without anti-NRP1 every other day through day +10. At day +40 following irradiation, 17% of the IgG-treated control mice remained alive. In contrast, 83% of irradiated mice treated with anti-NRP1 remained alive and well-appearing ($n = 12$/group, $p = 0.001$, Fig. 3f). These results suggest that anti-NRP1 treatment promoted the recovery of radioprotective HSPCs following TBI[54]. Irradiated mice treated with anti-NRP1 displayed increased hemoglobin concentrations at day +7, increased WBCs at day +10 (Fig. 3g) and substantially increased BM KSL cells at day +10 compared to irradiated control mice (Fig. 3h).

**NRP1 regulates vascular regeneration following chemotherapy.** In order to determine if NRP1 inhibition could promote BM vascular regeneration in mice following chemotherapy, we treated C57BL/6J mice with 5 fluorouracil (5FU), 250 mg/kg intravenously × 1, and then administered 10 μg anti-NRP1 or isotype every other day. 5FU treatment strongly induced Sema3a expression in BM ECs at 24 h (Fig. 4a). At day +7 following 5FU, we observed marked disruption of the BM vasculature in control mice; in contrast, mice treated with 5FU and anti-NRP1 displayed BM vascular reformation (Fig. 4b). Mice treated with 5FU and anti-NRP1 also displayed increased BM cell counts and increased percentages and numbers of VE-cad$^+$ BM ECs at day +7 compared to 5FU-treated controls (Fig. 4c, d). 5FU treatment increased percentages of apoptotic BM ECs at day +7 in control mice, but mice treated with 5FU followed by anti-NRP1 displayed decreased percentages of apoptotic BM ECs (Fig. 4e).

Importantly, mice treated with 5FU followed by anti-NRP1 demonstrated significantly increased PB WBCs and lymphocyte counts, increased percentages and numbers of BM KSL HSPCs, and increased BM CFCs at day +10 compared to mice treated

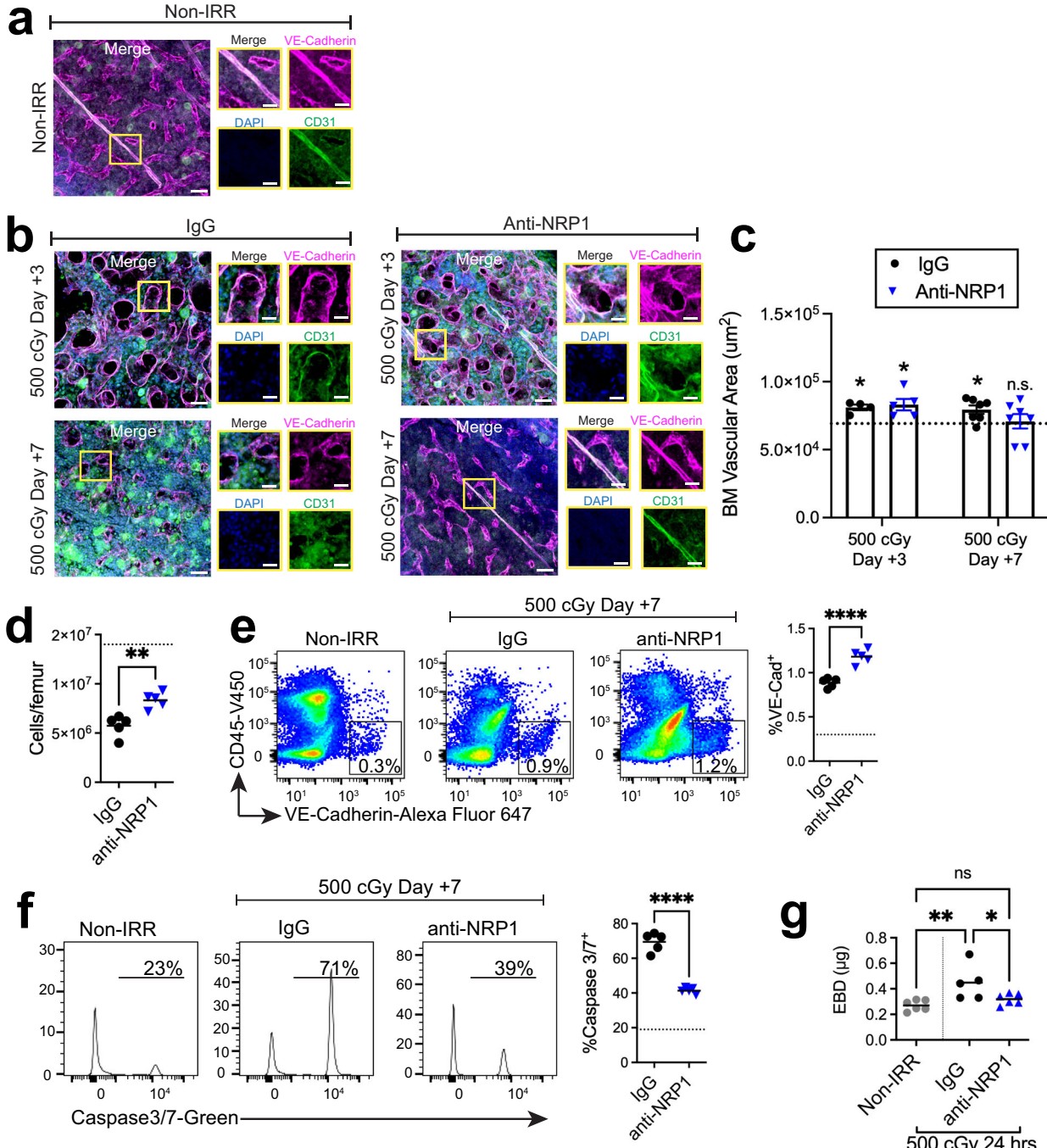

**Fig. 2 NRP1 inhibition accelerates BM vascular regeneration in vivo.** Representative confocal images of VE-cadherin+ (magenta) and CD31+ (green) BM vessels, and nuclei (blue) in femurs from **a** non-IRR mice and at **b** day +3 and day +7 following 500 cGy and treatment with IgG (10 μg/dose) or anti-NRP1 (10 μg/dose). Left image shows full view taken using 20× lens and right images display individual channels from magnified yellow box. Scale bar 50 μm; magnified view scale bar, 20 μm. **c** Quantification of VE-Cadherin vascular area from the images in **a**, **b**. Dotted line represents mean Non-IRR vascular area. Data presented as mean values +/− SEM. (n = 2 independent experiments, d3 IgG n = 4 fields of view, d3 anti-NRP1 n = 5 fields, d7 IgG n = 7 fields, d7 anti-NRP1 n = 7 fields; IgG d3: p = 0.016, anti-NRP1 d3: p = 0.0467, IgG d7: p = 0.0349, anti-NRP1 d7: p = 0.9694. **d** BM cell counts at day +7 following 500 cGy and treatment with IgG or anti-NRP1 (n = 5 mice/group, p = 0.0025). Dotted line shows cell counts of non-IRR BM. **e** At left, representative flow cytometric analysis of CD45−VE-cad+ BM ECs within BM lin− cells from the groups shown. At right, percentages of BM ECs at day +7 following 500 cGy. Dotted line represents %VE-cad+ ECs in non-IRR controls (n = 5 mice/group). **f** At left, representative histograms of % activated caspase 3/7+ cells within lin−CD45−VE-cad+ BM ECs at day +7 following 500 cGy. At right, mean % activated caspase 3/7+ BM ECs at day +7 following 500 cGy (n = 5/group). Dotted line represents % activated caspase 3/7+ ECs in non-IRR controls. **g** Levels of Evans Blue Dye (EBD) in the BM extracellular space at +24 h following 500 cGy and treatment with IgG or anti-NRP1 (n = 5–6 mice/group, non-IRR vs. IgG: p = 0.0024, IgG vs. anti-NRP1: p = 0.188, non-IRR vs. anti-NRP1 p = 0.2679). **d**–**f** Data assessed by Student's two-tailed t-test, **g** One-way ANOVA with Holm-Sidak's multiple comparison two-sided t-test, **c** Two-sided one-sample t-test and Wilcoxon test used to compare sample means to the mean of non-IRR samples, *p < 0.05, **p < 0.01, ****p < 0.0001. Source data provided as a Source Data file.

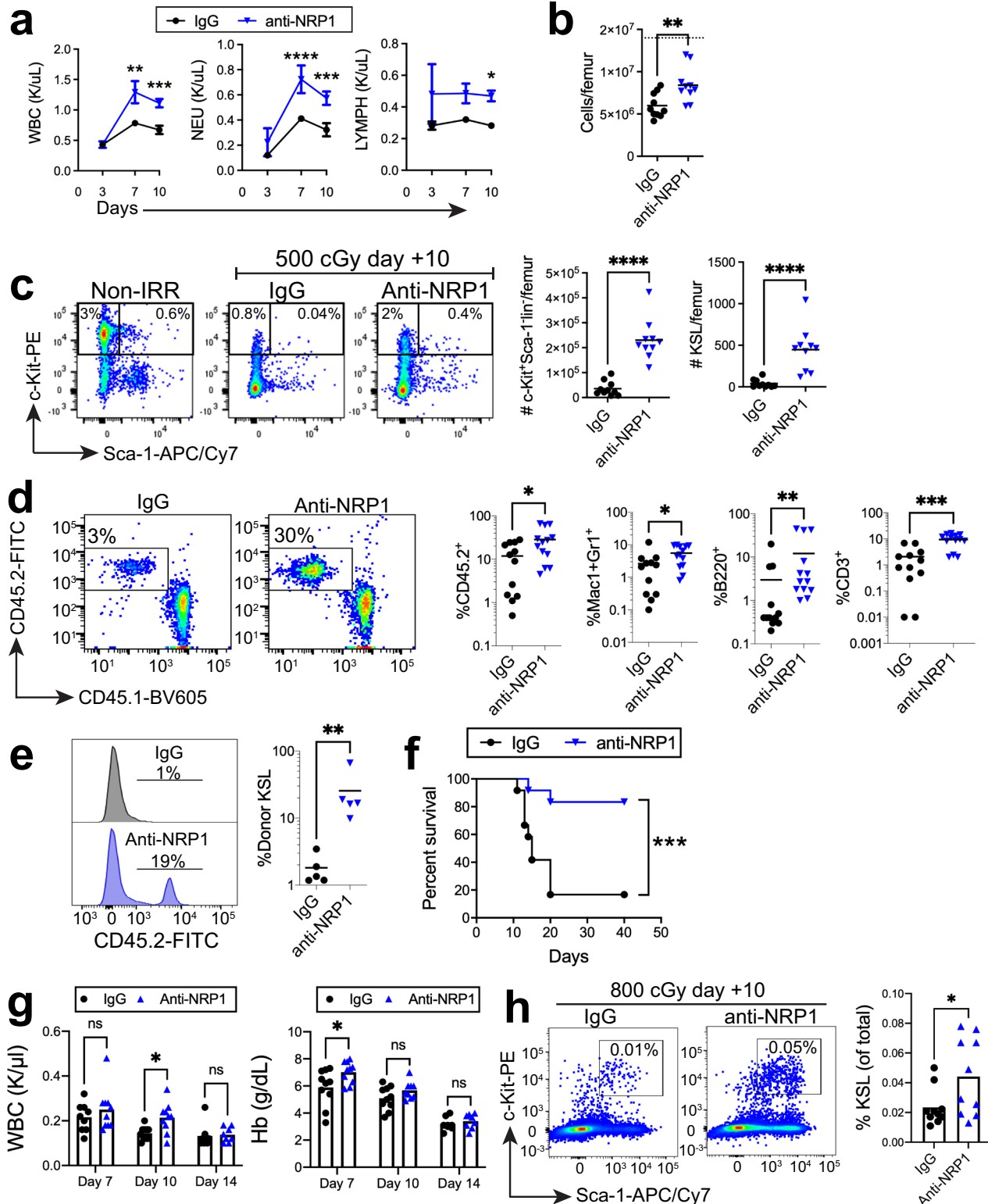

with 5FU followed by IgG, signifying early hematopoietic reconstitution in response to anti-NRP1 treatment (Fig. 4f–h).

**Nrp1 deletion in ECs accelerates BM vascular regeneration.** We next crossed *Cdh5-Cre-ERT2* mice (*Cdh5* encodes VE-cadherin; *Cdh5-Cre-ERT2* mice, developed by Ralf H. Adams Laboratory) with *Nrp1fl/fl* mice to generate *Cdh5-Cre-ERT2;Nrp1fl/fl* mice in which *Nrp1* deletion occurs in VE-cad+ ECs under the control of

tamoxifen treatment[55]. *Cdh5-Cre-ERT2;Nrp1fl/fl* mice displayed decreased expression of *Nrp1* in BM ECs compared to *Nrp1fl/fl* control mice following tamoxifen treatment (Fig. 5a). We observed no baseline differences in the BM vasculature or BM EC content in *Cdh5-Cre-ERT2;Nrp1fl/fl* mice compared to *Nrp1fl/fl* mice (Supplementary Fig. 5a, b). *Cdh5-Cre-ERT2;Nrp1fl/fl* mice also displayed no baseline changes in BM cell counts, PB complete blood counts, BM KSL cells or LT-HSCs compared to

**Fig. 3 NRP1 inhibition accelerates hematopoietic regeneration following TBI. a** PB WBC, neutrophil (NEU), and lymphocyte (LYMPH) counts at days +3, +7, and +10 following 500 cGy and treatment with IgG or anti-NRP1 ($n = 5$–10 mice/group, mean values +/− SEM; WBC: d7: $p = 0.0016$, d10 $p = 0.002$; NEU: d7: $p = 0.0089$, d10: $p = 0.0035$, LYMPH: d10: $p = 0.0482$). **b** BM cell counts at day +10 following 500 cGy and treatment with IgG or anti-NRP1 ($n = 5$–10 mice/group, $p = 0.0033$). Dotted line represents BM cell counts in non-IRR controls. **c** At left, flow cytometry of BM c-kit$^+$sca-1$^-$lin$^-$ progenitors and KSL cells at day +10 following 500 cGy; at right, numbers of BM c-kit$^+$sca-1$^-$lin$^-$ cells and KSL at day +10 ($n = 5$–10 mice/group, $p < 0.0001$). **d** At left, PB donor CD45.2$^+$ cell engraftment at 20 weeks in CD45.1$^+$ mice transplanted with BM cells collected at day +10 from 500 cGy—irradiated CD45.2$^+$ mice treated with IgG or anti-NRP1, along with CD45.1$^+$ BM competitor cells. At right, % total CD45.2$^+$ cells, CD45.2$^+$Mac1$^+$Gr1$^+$ myeloid cells, CD45.2$^+$B220$^+$ B cells, and CD45.2$^+$CD3$^+$ T cells at 20 weeks ($n = 12$–13 mice/group, total: $p = 0.0288$, myeloid: $p = 0.0266$, B cells: $p = 0.0051$, T cells: $p = 0.0001$). **e** At left, histograms of donor CD45.2$^+$KSL cells in the BM of transplanted CD45.1$^+$ mice at 20 weeks ($n = 5$ mice/group, $p = 0.0079$). **f** Percent survival of C57BL/6J mice following 800 cGy TBI and treatment with anti-NRP1 or IgG for 10 days (Log-rank test, $p = 0.001$, $n = 12$ mice/group). **g** PB WBCs and hemoglobin (Hb) at day +7, +10 and +14 from C57BL/6J mice following 800 cGy and treatment with anti-NRP1 or IgG ($n = 8$–10 mice/condition; WBC d7: $p = 0.4045$, WBC d10: $p = 0.0415$, WBC d14: $p = 0.8714$; Hb d7: $p = 0.0154$, Hb d10: $p = 0.2741$, Hb d14: $p = 0.6216$). **h** At left, flow cytometry of BM KSL cells 10 days following 800 cGy and treatment with anti-NRP1 or IgG; at right, %KSL cells ($n = 10$ mice/condition; $p = 0.0358$). **a** Two-way ANOVA with Holm Sidak's multiple comparison two-sided t-test. **b, c** Two-sided t-test, **d, e** Two-sided Mann–Whitney test, **f** Two-sided log-rank test, **g** Two-way ANOVA with Holm-Sidak's multiple comparison two-sided t-tests, and **h** Student's two-sided unpaired t-test. $^*p < 0.05$, $^{**}p < 0.01$, $^{***}p < 0.001$, $^{****}p < 0.0001$. Source data are provided as a Source Data file.

control *Nrp1*$^{fl/fl}$ mice (Supplementary Fig. 5c–e). However, at day +7 following 500 cGy TBI, *Cdh5-Cre-ERT2;Nrp1*$^{fl/fl}$ mice demonstrated accelerated recovery of the BM vasculature and increased BM cell counts compared to irradiated *Nrp1*$^{fl/fl}$ controls (Fig. 5b, c). *Cdh5-Cre-ERT2;Nrp1*$^{fl/fl}$ mice also showed increased percentages and numbers of BM ECs and decreased percentages of apoptotic and necrotic BM ECs compared to *Nrp1*$^{fl/fl}$ mice at day +10 following TBI (Fig. 5d–f).

At day +10 following TBI, *Cdh5-Cre-ERT2;Nrp1*$^{fl/fl}$ mice also displayed increased PB WBCs, neutrophils and lymphocytes, as well as increased percentages and numbers of BM ckit$^+$sca-1$^-$lin$^-$ myeloid progenitors, KSL cells and BM CFCs compared to irradiated *Nrp1*$^{fl/fl}$ mice (Fig. 5g–i and Supplementary Fig. 5f). However, competitive repopulation assays showed no significant differences in total donor engraftment through 20 weeks in congenic recipient mice transplanted with donor BM cells collected at day +10 from 500 cGy TBI-irradiated *Cdh5-Cre-ERT2;Nrp1*$^{fl/fl}$ mice versus irradiated *Nrp1*$^{fl/fl}$ mice (Supplementary Fig. 5g).

**Sema3a deletion in ECs phenocopies Nrp1 deletion.** We next tested whether EC-specific deletion of *Sema3a*, which encodes the ligand for NRP1, could recapitulate the effects of EC-specific *Nrp1* deletion on BM vascular regeneration. We crossed *Cdh5-Cre-ERT2* mice with *Sema3a*$^{fl/fl}$ mice (RIKEN) to generate *Cdh5-Cre-ERT2;Sema3a*$^{fl/fl}$ mice[56]. Tamoxifen-treated *Cdh5-Cre-ERT2;Sema3a*$^{fl/fl}$ mice demonstrated loss of *Sema3a* expression in BM ECs (Supplementary Fig. 6a). We observed no differences in the BM vasculature or vascular area at baseline between *Cdh5-Cre-ERT2;Sema3a*$^{fl/fl}$ mice and *Sema3a*$^{fl/fl}$ mice, although percentages of CD31$^+$ BM ECs were moderately increased in *Cdh5-Cre-ERT2;Sema3a*$^{fl/fl}$ mice (Supplementary Fig. 6b, c). BM cell counts, PB complete blood counts and percentages of BM KSL cells and LT-HSCs were not different in *Cdh5-Cre-ERT2;Sema3a*$^{fl/fl}$ mice at baseline compared to *Sema3a*$^{fl/fl}$ controls (Supplementary Fig. 6d–f). However, at day +7 following 500 cGy TBI, *Cdh5-CreERT2;Sema3a*$^{fl/fl}$ mice displayed enhanced BM vascular recovery and increased BM cell counts compared to *Sema3a*$^{fl/fl}$ mice (Supplementary Fig. 7a, b). Percentages of CD31$^+$ BM ECs were increased and percentages of apoptotic and necrotic CD31$^+$ BM ECs were decreased in *Cdh5-CreERT2;Sema3a*$^{fl/fl}$ mice compared to *Sema3a*$^{fl/fl}$ mice following TBI (Supplementary Fig. 7c–e). At day +10 following 500 cGy TBI, *Cdh5-Cre-ERT2;Sema3a*$^{fl/fl}$ mice also displayed increased PB WBCs and neutrophils, increased percentages of BM KSL cells and increased ckit$^+$sca-1$^-$lin$^-$ myeloid progenitor cells compared to *Sema3a*$^{fl/fl}$ mice (Supplementary Fig. 7f, g). In

summary, EC-specific deletion of *Sema3a* phenocopied the effects of EC-specific deletion of *Nrp1* in irradiated mice.

**NRP1 control of BM vascular regeneration is VEGF-independent.** In addition to its role as a receptor for SEMA3A, NRP1 can serve as a receptor for vascular endothelial growth factor-A 165 (VEGF-A165)[57,58]. VEGF-A 165 binding to NRP1 induces the formation of an NRP1–VEGFR2 complex, thereby promoting VEGFR-2 intracellular signaling[57–59]. Via this NRP1–VEGFR2 co-receptor mechanism, NRP1 regulates vascular angiogenesis and arteriogenesis[60,61]. In order to determine whether anti-NRP1 antibody treatment mediated effects on the BM vasculature via modulation of VEGF–NRP1 signaling in BM ECs, we utilized *Nrp1*$^{VEGF-}$ mice (courtesy of Dr. Chenghua Gu, Harvard), which have a point mutation in the endogenous *Nrp1* locus that selectively abolishes VEGF–NRP1 binding[62]. *Nrp1*$^{VEGF-}$ mice demonstrated no baseline differences in percentages of VE-cad$^+$ BM ECs compared to control mice (Supplementary Fig. 8a). We next irradiated *Nrp1*$^{VEGF-}$ mice and control mice with 500 cGy TBI and treated with or without anti-NRP1 antibody every other day for 7 days. Irradiated *Nrp1*$^{VEGF-}$ mice treated with isotype antibody displayed increased CD31$^+$ BM EC apoptosis, comparable to that observed in irradiated controls (Fig. 6a). Irradiated *Nrp1*$^{VEGF-}$ mice treated with anti-NRP1 demonstrated decreased CD31$^+$ BM EC apoptosis compared to *Nrp1*$^{VEGF-}$ mice treated with isotype (Fig. 6a). These results suggest that anti-NRP1 treatment promotes BM vascular recovery via a mechanism independent from VEGF–NRP1 signaling. We next transplanted congenic CD45.1$^+$ *B6.SJL* mice with equal doses of BM cells collected at day +10 from irradiated (45.2+) *Nrp1*$^{VEGF-}$ mice treated with anti-NRP1 antibody or isotype. Recipient mice transplanted with BM cells from irradiated *Nrp1*$^{VEGF-}$ mice treated with anti-NRP1 antibody displayed approximately 1 log increased total donor hematopoietic cell engraftment and increased donor myeloid and T cell chimerism at 20 weeks of post-transplant compared to recipient mice transplanted with equal doses of BM cells from irradiated, isotype- treated *Nrp1*$^{VEGF-}$ mice (Fig. 6b). These results suggest that anti-NRP1 treatment promotes hematopoietic regeneration independently from VEGF–NRP1 signaling.

**SEMA3A promotes BM EC apoptosis via Cdk5 and p53.** In the nervous system, SEMA3A promotes neurite growth cone collapse through activation of Cyclin dependent kinase 5 (Cdk5), which was suggested to occur via the NRP1 co-receptor, Plexin A2[63,64]. Subsequent studies have shown that SEMA3A primarily signals through NRP1 co-receptor, Plexin A4[65–67]. We detected a

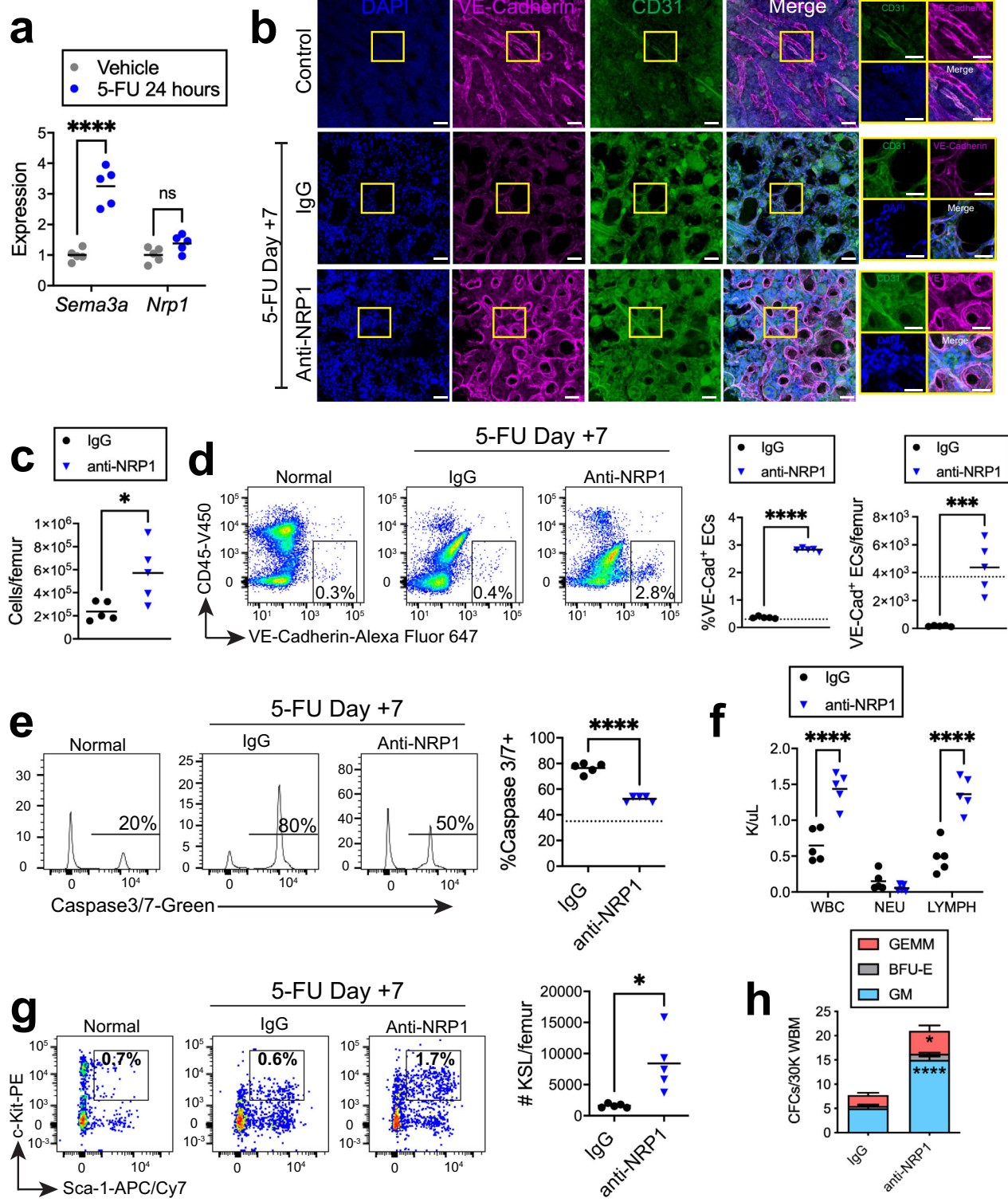

significant increase in Cdk5 phosphorylation in BM ECs at 1 h following irradiation with 800 cGy (Fig. 6c). Treatment with SEMA3A further increased Cdk5 phosphorylation in BM ECs, whereas anti-NRP1 treatment suppressed TBI-induced and SEMA3A-induced phosphorylation of Cdk5 in BM ECs (Fig. 6c). Irradiation and SEMA3A treatment both increased CD31+ BM EC apoptosis in vitro at +24 h, whereas treatment with anti-NRP1 abrogated these effects of irradiation and SEMA3A (Fig. 6d). Furthermore, treatment with roscovitine, a Cdk5 inhibitor[68], abolished SEMA3A-mediated induction of BM EC

apoptosis in vitro, suggesting that SEMA3A-mediated induction of BM EC apoptosis was dependent on Cdk5 (Fig. 6d).

In neurons, Cdk5 phosphorylates the tumor suppressor, p53, causing stabilization of p53 and increased neuronal cell death[69,70]. We detected increased phosphorylation of p53 in BM ECs in response to irradiation and a further increase with SEMA3A treatment (Fig. 6e). Treatment with anti-NRP1 abolished SEMA3A-mediated phosphorylation of p53 in irradiated BM ECs (Fig. 6e). Similarly, treatment with roscovitine suppressed p53 phosphorylation in BM ECs in response to

**Fig. 4 NRP1 inhibition accelerates BM vascular and hematopoietic regeneration following chemotherapy. a** qRT-PCR analysis of *Sema3a* and *Nrp1* expression in BM ECs from mice at +24 h following 5-FU chemotherapy. For each gene, expression is normalized to vehicle treatment (n = 5/condition; *Sema3a*: p < 0.0001, *Nrp1*: p = 0.1291). **b** Representative images of VE-cadherin⁺ (magenta)⸴ CD31⁺ (green) BM vessels in femur sections from untreated control mice and at day +7 following 5-FU chemotherapy and treatment with anti-NRP1 or IgG. Nuclei are stained with DAPI (blue). Scale bar, 50 μm; magnified view scale bar, 20 μm. **c** BM cell counts at day +7 following 5-FU chemotherapy and treatment with anti-NRP1 or IgG (n = 5 mice/group, p = 0.0219). **d** At left, representative flow cytometric analysis of CD45⁻VE-cad⁺ BM ECs within BM lin⁻ cells from control mice and at day +7 following 5-FU chemotherapy and anti-NRP1 or IgG treatment. At right, percentages and numbers of BM ECs at day +7 following 5-FU chemotherapy and the treatments shown. Dotted lines represent numbers in control mice (n = 5 mice/group; % BM ECs p < .0001; number BM ECs p = 0.0008). **e** At left, representative histograms show % activated caspase 3/7⁺ cells within lin⁻CD45⁻VE-cad⁺ BM ECs at day +7 following 5-FU. At right, % activated caspase 3/7⁺ BM ECs at day +7 (n = 5/group; p < .0001). Dotted line represents percentages of activated caspase 3/7⁺ cells within lin⁻CD45⁻VE-cad⁺ BM ECs in non-IRR controls. **f** PB WBC, NEU, and LYMPHs at day +7 following 5-FU and the treatments shown (n = 5 mice/group; p < 0.0001). **g** At left, flow cytometry of BM KSL cells at day +7 following 5-FU and the treatments shown. At right, % BM KSL cells (n = 5 mice/group; p = 0.011). **h** Colony forming unit quantification from 30,000 BM cells isolated from mice at day +7 following 5-FU and IgG or anti-NRP1 treatment (n = 4 replicates, mean values +/− SEM; GM: p < 0.0001; GEMM: p = 0.0201). Statistics show GM and GEMM comparisons. **a**, **f**, **h** Two-way ANOVA followed by Holm-Sidak's multiple comparison two-sided t-tests; **c**, **d**, **g** Student's unpaired two-sided t-test. *p < 0.05, ***p < 0.001, ****p < 0.0001. Source data provided as a Source Data file.

irradiation and SEMA3A, suggesting that SEMA3A-mediated activation of p53 was dependent on Cdk5 (Fig. 6e). In order to determine if the anti-apoptotic effects of anti-NRP1 on BM ECs were dependent on suppression of p53, we irradiated BM ECs from *p53⁻/⁻* mice and *p53⁺/⁺* mice and treated with or without anti-NRP1. Anti-NRP1 blocked Caspase 3/7 activation in *p53⁺/⁺* BM ECs in response to 800 cGy, but had no effect on Caspase 3/7 activation in *p53⁻/⁻* BM ECs (Fig. 6f). Therefore, NRP1-mediated induction of BM EC apoptosis following irradiation was dependent on p53 activation.

Following irradiation, p53 drives programmed cell death by promoting the expression of pro-apoptotic genes, including p53-upregulated modulator of apoptosis (*Puma*), *Noxa*, and Bcl2-associated X protein (*Bax*)[71]. We observed no significant increase in expression of *Bax* or *Bak1* in BM ECs following 800 cGy, with or without SEMA3A treatment, although expression of *Bad* increased in response to 800 cGy with no additional response to SEMA3A (Supplementary Fig. 8b). However, expression of *Puma* increased in BM ECs following irradiation and further increased in response to SEMA3A treatment (Fig. 6g). Treatment with anti-NRP1 or roscovitine suppressed *Puma* expression in BM ECs that otherwise occurred in response to irradiation and SEMA3A treatment (Fig. 6g). These data suggest that SEMA3A-mediated induction of *PUMA* in BM ECs is dependent on NRP1 and activation of Cdk5.

**BM ECs secrete R spondin 2 in response to NRP1 inhibition.** In order to elucidate the mechanisms through which BM ECs promote hematopoietic regeneration in response to anti-NRP1 treatment, we isolated CD45⁻CD31⁺ BM ECs from the femurs of adult *C57BL/6J* mice at 72 h following 500 cGy TBI and treatment with anti-NRP1 or IgG. RNA sequence analysis revealed significant changes in gene expression in BM ECs from anti-NRP1-treated mice compared to controls (GEO database, accession number GSE149776). Ingenuity pathway analysis demonstrated alterations in several pathways in BM ECs in response to anti-NRP1, including organismal survival, hematological system development and function, hematopoiesis, cardiovascular disease and hematologic disease pathways (Fig. 7a). We then specifically examined the expression of genes that encode secreted proteins (Fig. 7b)[72]. We detected increased expression of *Kitl* and *Cxcl12*, which encode proteins that are critical for HSC maintenance (Fig. 7b)[3–5], in regenerating BM ECs following anti-NRP1 treatment. The most highly overexpressed gene in regenerating BM ECs in response to NRP1 inhibition was *Rspo2*, which encodes R spondin 2, a Wnt signal amplifying protein that is not known to regulate adult hematopoiesis[73]. ELISA demonstrated

that BM ECs increased secretion of R spondin 2 protein more than 20-fold in response to anti-NRP1 treatment (Fig. 7c). We also examined the cell surface expression of candidate receptors for R spondin 2 on BM CD34⁻KSL HSCs, which would include the leucine-rich repeat-containing G-protein coupled receptor 5 (LGR5)[74,75], before and following 500 cGy TBI. We did not detect significant surface expression of LGR5 on BM HSCs at baseline or following TBI (Supplementary Fig. 9a–d). Gene expression analysis also did not reveal significant upregulation of other Lgr genes, *Lgr4* and *Lgr5*, or other R spondin receptor genes, *Fz8* or *Lrp6* on BM HSCs following 500 cGy TBI (Supplementary Fig. 9e).

Since regenerating BM ECs upregulate expression and secretion of R spondin 2 in response to anti-NRP1 treatment, we tested whether hematopoietic regeneration in irradiated mice in response to NRP1 inhibition was dependent on R spondin 2. We irradiated adult *C57BL/6J* mice with 500 cGy TBI and treated with anti-NRP1 every other day for 10 days, with and without systemic administration of 10 μg anti-R spondin 2 antibody. Anti-NRP1-treated mice displayed significantly increased PB WBCs and neutrophils at day +10 following TBI compared to controls (Fig. 7d). Conversely, mice treated with anti-NRP1 and anti-R spondin 2 showed no increase in PB WBCs or neutrophils compared to controls (Fig. 7d). Irradiated mice treated with anti-NRP1 also demonstrated enhanced recovery of BM LT-HSCs and KSL cells at day +10 compared to irradiated controls; concurrent treatment with anti-R spondin 2 blocked these effects (Fig. 7e, f). Irradiated mice treated with anti-NRP1 also demonstrated increased recovery of BM CFCs and multipotent colony forming unit—granulocyte, erythroid, megakaryocyte, macrophage (CFU-GEMMs) compared to irradiated controls at day +10, and administration of anti-R spondin 2 abrogated these effects (Fig. 7g). Since R spondin proteins have been shown to amplify canonical Wnt signaling[76], we tested whether R spondin 2 treatment alone or in combination with Wnt3a caused direct effects on BM hematopoietic progenitor cells following 300 cGy irradiation. Treatment of irradiated BM c-kit⁺lin⁻ progenitor cells with Wnt3a alone or R spondin 2 alone for 72 h did not increase total CFC recovery or CFU-GEMM recovery compared to irradiated control cultures (Fig. 7h and Supplementary Fig. 9f). However, the combination of R spondin 2 and Wnt3a significantly increased total CFC recovery and markedly increased recovery of multipotent CFU-GEMMs compared to irradiated control hematopoietic progenitor cells (Fig. 7h and Supplementary Fig. 9f). These data suggest that R spondin 2-mediated direct effects on hematopoietic progenitor regeneration require concomitant Wnt ligand-mediated activation.

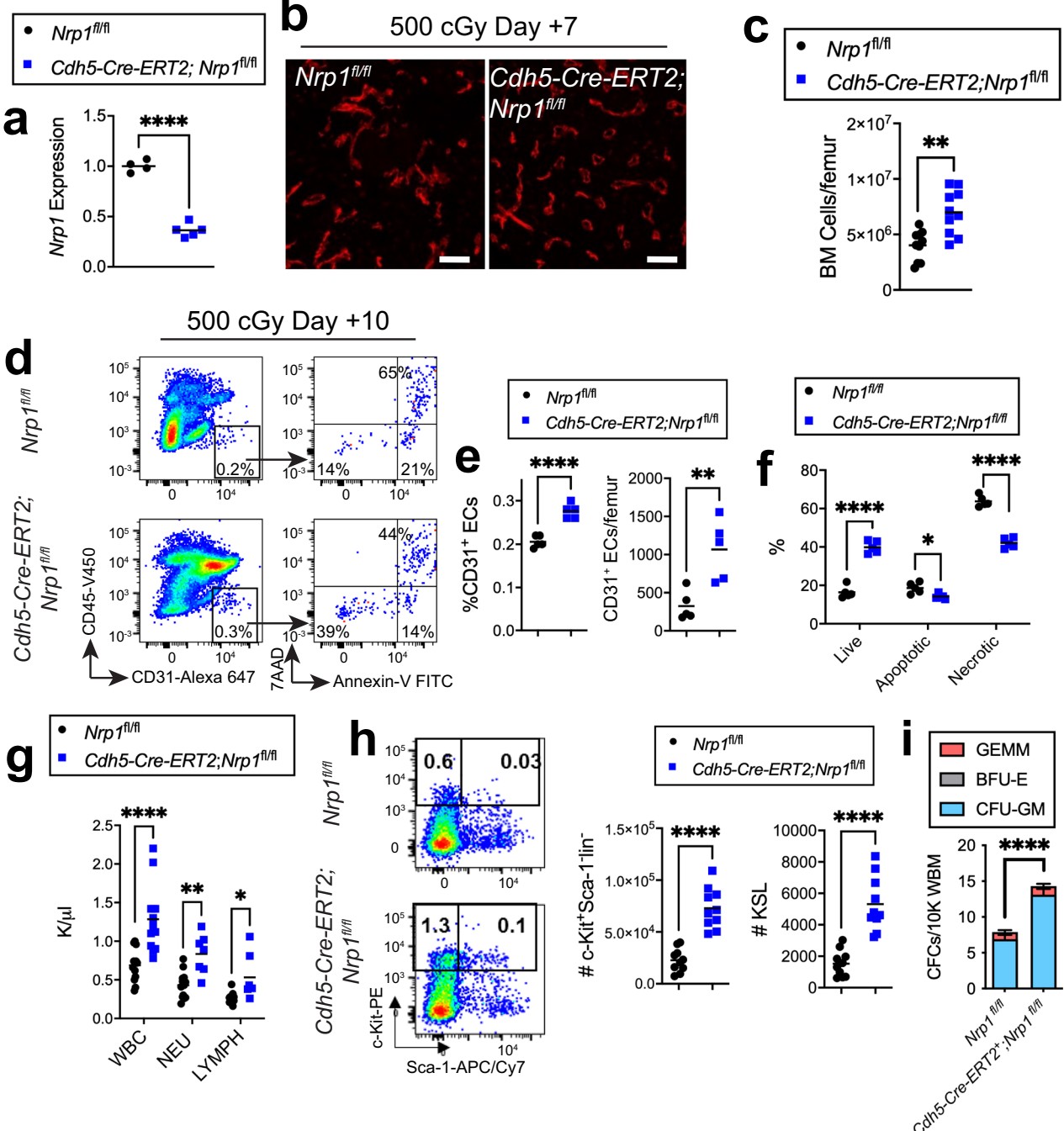

**Fig. 5 EC-specific deletion of *Nrp1* accelerates BM vascular and hematopoietic regeneration. a** qRT-PCR analysis for *Nrp1* expression in VE-cadherin⁺ BM ECs from *Nrp1^fl/fl* mice and *Cdh5-Cre-ERT2;Nrp1^fl/fl* mice following tamoxifen treatment (*n* = 4–5 mice/group, *p* < 0.0001). *Nrp1* expression was normalized to *Gapdh* levels and then replicates were normalized relative to the *Nrp1^fl/fl* group. **b** Representative microscopic images of VE-cad⁺ BM vessels (red) in femur sections from *Nrp1^fl/fl* mice and *Cdh5-Cre-ERT2;Nrp1^fl/fl* mice at day +7 following 500 cGy TBI. Scale bar, 100 μm. **c** BM cell counts in *Nrp1^fl/fl* mice and *Cdh5-Cre-ERT2;Nrp1^fl/fl* mice at day +10 following 500 cGy (*n* = 10 mice/group; *p* = 0.001). **d** Representative flow cytometric analysis of CD45⁻CD31⁺ BM ECs within BM lin⁻ cells from *Nrp1^fl/fl* mice and *Cdh5-Cre-ERT2;Nrp1^fl/fl* mice are shown at day +10. At right, representative percentages of Annexin⁺7AAD⁻ and Annexin⁺7AAD⁺ cells within the BM EC population are shown. **e** At left, mean percentages of CD31⁺ BM ECs within the BM lin⁻ population at day +10 and at right, numbers of femoral CD31⁺ BM ECs at day +10 following 500 cGy TBI (*n* = 5 mice/group; %BM ECs: *p* < 0.0001; number BM ECs: *p* = 0.0056). **f** Mean percentages of live, apoptotic, and necrotic BM ECs at day +10 following 500 cGy in the groups shown (*n* = 5 mice/group; *p* < 0.0001). **g** PB WBC, NEU, and LYMPHS at day +10 following 500 cGy TBI in *Nrp1^fl/fl* mice and *Cdh5-Cre-ERT2;Nrp1^fl/fl* mice (*n* = 10 mice/group; WBC: *p* < 0.0001; NEU: *p* = 0.0014; LYMPH: *p* = 0.0327). **h** At left, representative flow cytometric analysis of BM c-kit⁺sca1⁻lin⁻ progenitors and KSL stem/progenitor cells at day +10 following 500 cGy TBI. At right, number of c-kit⁺sca1⁻lin⁻ progenitors and KSL cells at day +10 in the groups shown (*n* = 10 mice/group, *p* < 0.0001). **i** CFC quantification from 10,000 BM cells isolated from mice at day +10 following IR (*n* = 10 replicates, data are presented as mean values +/− SEM, *p* < 0.0001). **a**, **c**, **e**, **h** Student's unpaired two-sided *t*-test, **f**, **g**, **i** Two-way ANOVA with Holm-Sidak's multiple comparison two-sided *t*-tests. *\*p* < 0.05, \*\**p* < 0.01, \*\*\*\**p* < 0.0001. Source data are provided as a Source Data file.

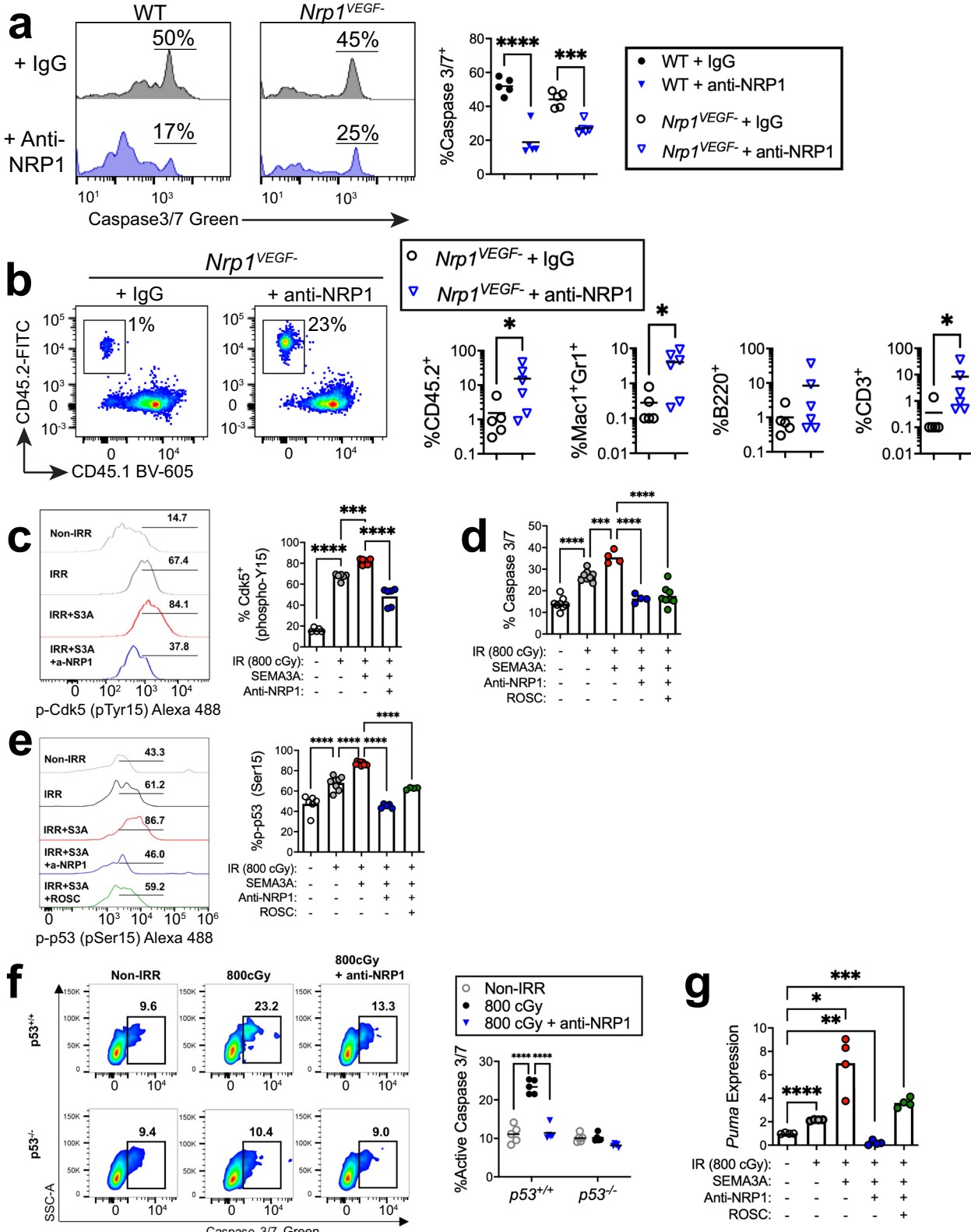

## Discussion

In addition to cell-intrinsic mechanisms[77–79], signaling from the BM microenvironment plays an important role in regulating hematopoietic regeneration following myeloablative insults[27,80–83]. BM ECs are indispensable in this process as deletion of VEGFR2[+] BM ECs has been shown to prevent HSC regeneration and hematologic recovery after TBI or chemotherapy[19]. Importantly,

myeloablation increases BM EC death and severely disrupts the BM vascular niche in a manner persistent for several weeks, thereby contributing to hematopoietic suppression[13,24,28,84]. Therefore, strategies to accelerate BM vascular regeneration could have profound therapeutic benefits toward accelerating hematopoietic reconstitution in patients following myeloablation. However, the mechanisms governing BM vascular regeneration following

**Fig. 6 SEMA3A promotes BM EC apoptosis via activation of Cdk5 and p53. a** (Left) flow cytometry of activated caspase 3/7$^+$ BM ECs in wild-type (WT) and $Nrp1^{VEGF-}$ mice at day +5 following 500 cGy and the treatments shown; (Right) % activated caspase 3/7$^+$ BM ECs ($n = 5$/group; ****$p < 0.0001$; ***$p = 0.0003$). **b** (Left) Donor CD45.2$^+$ cell engraftment at 20 weeks in the PB of CD45.1$^+$ mice competitively transplanted with BM cells from 500 cGy—irradiated $Nrp1^{VEGF-}$ mice treated with anti-NRP1 or IgG; (Right) % donor CD45.2$^+$, Mac1$^+$/Gr1$^+$, B220$^+$, and CD3$^+$ cells at 20 weeks ($n = 5$–6/group; total: $p = 0.0346$; myeloid: $p = 0.0325$; T cells: $p = 0.0195$). **c** (Left) Histograms of p-Cdk5$^+$ BM ECs at 1 h following 800 cGy and the treatments shown; (Right) %p-p53$^+$ BM ECs. Horizontal lines show gating for p-Cdk5$^+$ cells ($n = 5$ replicates, Non-IRR; $n = 6$ replicates for IRR, IRR + SEMA3A, IRR + SEMA3A + anti-NRP1; non-IRR vs. IRR: $p < 0.0001$; IRR vs. IRR + SEMA3A: $p = 0.0005$; IRR vs. IRR + SEMA3A + Anti-NRP1 $p < 0.0001$). **d** % activated caspase 3/7$^+$ BM ECs at 24 h following 800 cGy and treatment with SEMA3A, SEMA3A + anti-NRP1 or SEMA3A + roscovitine (ROSC), 10 ng/ml ($n = 4$–8 replicates; non-IRR vs. IRR: $p < 0.0001$; IRR vs. IRR + SEMA3A: $p = 0.0007$; IRR vs. IRR + SEMA3A + Anti-NRP1 $p < 0.0001$; IRR vs. IRR + SEMA3A + Anti-NRP1 + ROSC $p < 0.0001$). **e** At left, histograms of p-p53$^+$ BM ECs at 1 h following 800 cGy and treatment with SEMA3A, SEMA3A + anti-NRP1, or SEMA3A + ROSC. At right, % p-p53$^+$ BM ECs are shown. Black horizontal lines show gating for p-p53$^+$ cells ($n = 6$ replicates, Non-IRR; $n = 9$ replicates for IRR and IRR + SEMA3A; $n = 5$ replicates for IRR + SEMA3A + anti-NRP1; $n = 4$ replicates for IRR + SEMA3A + ROSC; $p < 0.0001$ for all comparisons). **f** At left, flow cytometry for activated caspase 3/7$^+$ BM ECs from $p53^{-/-}$ and $p53^{+/+}$ mice at 24 h following irradiation with 800 cGy and the treatments shown. At right, % activated caspase 3/7$^+$ BM ECs are shown ($n = 5$ replicates/group, $p < 0.0001$). **g** $Puma$ expression in BM ECs at 6 h following 800 cGy and the treatments shown ($n = 4$/condition; ****$p < 0.0001$, *$p = 0.0142$, **$p = 0.0025$, ***$p = 0.0009$). **a, c, d, e** One-way ANOVA followed by Holm-Sidak's two-sided unpaired $t$-tests, **f** Two-way ANOVA followed by Holm-Sidak's two-sided unpaired $t$-test, **g** Brown–Forsythe ANOVA followed by two-sided unpaired $t$-tests with Welch's correction, **b** Two-sided Mann–Whitney test. *$p < 0.05$, **$p < 0.01$, ***$p < 0.001$, ****$p < 0.0001$. Source Data file provided.

myelotoxicity are not well understood, and therapeutic approaches to accelerate BM vascular regeneration have yet to be developed. Here, we have demonstrated that BM ECs secrete SEMA3A in response to myeloablation and SEMA3A suppresses BM vascular regeneration via signaling through its receptor, NRP1. While SEMA3A has been suggested to have anti-angiogenic effects in models of multiple myeloma[85,86], our study reveals that BM ECs in normal mice markedly upregulate secretion of SEMA3A in response to TBI, thereby potentiating BM EC apoptosis and BM vascular dysfunction. We hypothesize that systemic administration of recombinant SEMA3A to irradiated mice did not worsen BM vascular damage or hematopoietic toxicities because high levels of endogenous SEMA3A in the BM of irradiated mice obscured potential effects of exogenously administered SEMA3A. Importantly, our studies suggest that the primary cellular source of SEMA3A is CD31$^+$Sca-1$^-$ sinusoidal BM ECs, which is in keeping with single cell gene expression analysis of sinusoidal BM ECs performed by Baryawno et al. and Tikhanova et al.[87–89].

These results provide a mechanism for therapeutic targeting of BM ECs to promote both BM vascular regeneration and hematopoietic recovery following myelosuppression. We demonstrate that antibody-mediated blockade of NRP1 or EC-specific deletion of $Nrp1$ or its ligand, $Sema3a$, promotes BM vascular regeneration and concordantly drives regeneration of the HSC pool and the hematopoietic system in myeloablated mice. Notably, anti-NRP1 treatment did not prevent TBI-induced BM vascular damage at day +3, but rather accelerated recovery of the BM sinusoidal architecture thereafter. Furthermore, BM vascular regeneration that occurred in response to anti-NRP1 treatment was not related to modulation of VEGF-A–NRP1 signaling since anti-NRP1 treatment produced identical regenerative effects in $Nrp1^{VEGF-}$ mice and control mice. These data suggest that SEMA3A–NRP1 signaling in ECs is specifically responsible for BM vascular damage following myelosuppression and this mechanism can be therapeutically inhibited to drive BM vascular regeneration and hematopoietic reconstitution in vivo. These results also raise the possibility that SEMA3A–NRP1 signaling could be targeted to promote vascular regeneration in other tissues.

In addition to elucidating the role of SEMA3A–NRP1 signaling in regulating BM vascular regeneration following myeloablation, we describe a mechanism through which regenerating BM ECs promote hematopoietic regeneration in vivo. In response to NRP1 inhibition, regenerating BM ECs increased expression and secretion of R spondin 2[73,76]. Furthermore, we showed that

R spondin 2 contributes to HSC regeneration and hematologic recovery that occurs in response to NRP1 inhibition. R spondins 1 and 3 have been shown to facilitate mesoderm transition of human pluripotent stem cells[90] and R spondin 1 has been implicated in regulating HSC specification during development[73], but R spondins are not known to regulate adult hematopoiesis. In the intestine, R spondin proteins (1–4) bind LGRs 4, 5, and 6, and function as Wnt signal enhancers in adult intestinal stem cells (ISCs)[76]. R spondins act by neutralizing Rnf43 and Znrf3, transmembrane E3 ligases that remove Wnt receptors from the epithelial stem cell surface, thereby amplifying canonical Wnt signaling[76,91,92]. In the intestinal crypt, R spondins and Wnt ligands are essential for maintenance of LGR5$^+$ ISCs[75]. However, Wnt ligands are insufficient to mediate ISC self-renewal, which is conferred by the action of R spondins, with a requirement for Wnt ligand—maintained expression of R spondin receptors[75,93]. R spondins have also been shown to amplify Wnt signaling via binding to LRP6 and blockade of Dkk1 activity[94], and through binding to Frizzled 8 receptor, causing direct amplification of Wnt signaling[95]. Following 500 cGy TBI, we did not detect upregulation in expression of $Lgr$ genes, $Lrp6$, or $Fzd8$ in BM HSCs, but R spondins 2 and 3 have been shown to promote Wnt signal activation in the absence of LGR proteins via engagement with heparan sulfate proteoglycans on the cell surface[96,97]. In future studies, we will focus on elucidation of the receptors responsible for mediating R spondin 2 effects on hematopoietic regeneration and whether such effects occur directly on HSCs or indirectly via cross-talk with BM niche cells.

In summary, SEMA3A–NRP1 signaling in BM ECs regulates the BM vascular response to myelosuppression and, in turn, hematopoietic reconstitution. Targeted inhibition of NRP1 promotes BM EC survival and vascular regeneration following TBI or chemotherapy and accelerates hematopoietic reconstitution in vivo. Regenerating BM ECs augment hematopoietic regeneration in response to NRP1 inhibition, at least in part, via secretion of R spondin 2. These results suggest that the SEMA3A–NRP1 signaling pathway in BM ECs can be therapeutically targeted to promote regeneration of the BM vascular niche and the hematopoietic system following myelosuppression.

## Methods

All research performed in this study complies with all ethical regulations and was approved by the UCLA Animal Research Committee (ARC #2014-021-13P, principal investigator: JPC) and the Cedars Sinai Medical Center Animal Use Committee (IACUC009617).

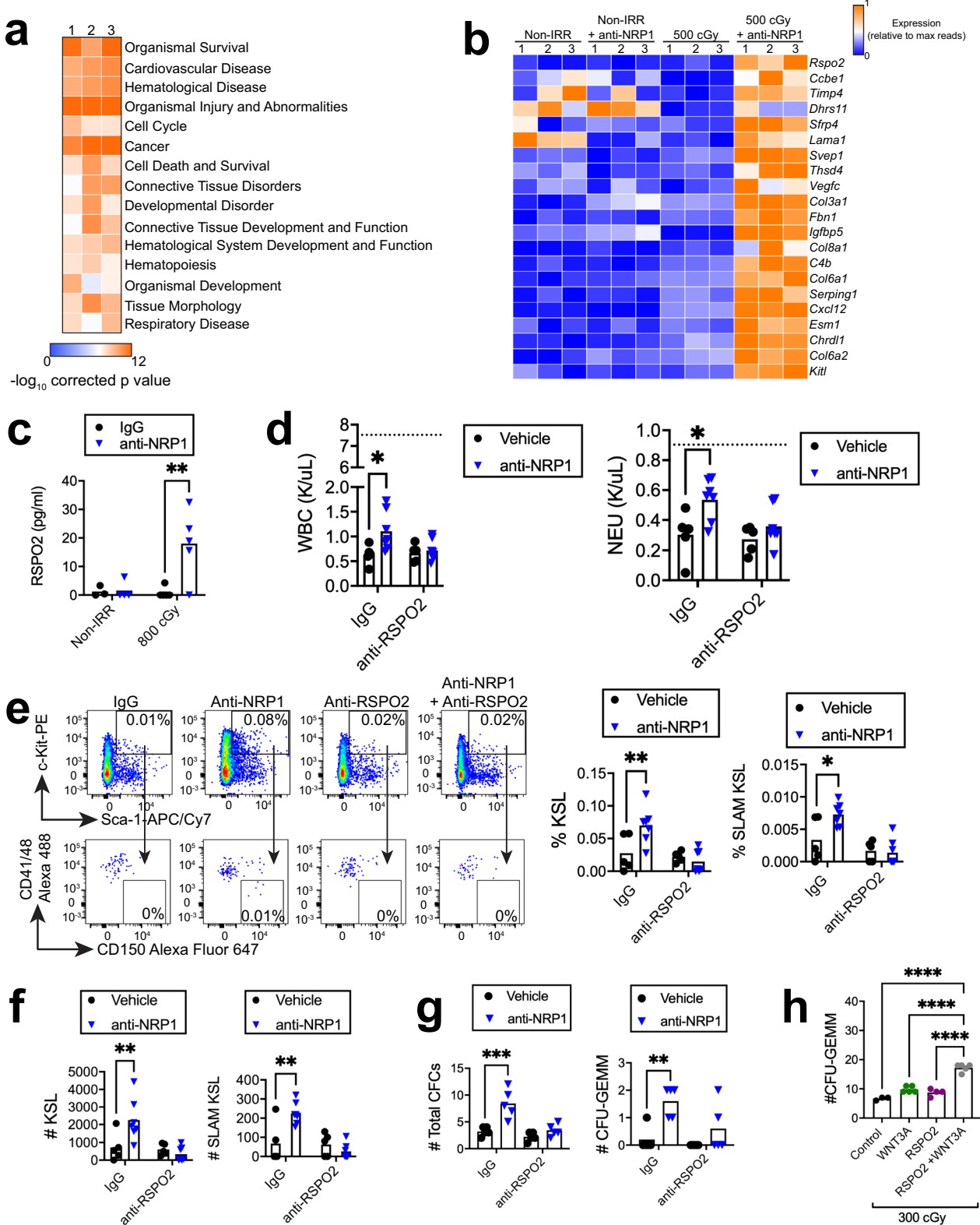

**Mice**. All animal procedures were performed in accordance with UCLA Animal Research Committee requirements and Cedars Sinai Medical Center Animal Use Committee requirements via the approved protocols described above. All mice were maintained in the UCLA Radiation Oncology vivarium, which is a barrier facility specifically designed for the care of mice following radiation treatment. Animals were housed in a clean barrier facility with room temperature conditions (20–26 °C), relative humidity 30–70%, with 12-h dark/light cycles. Experimental mice were separated by sex and housed 4–5 mice per cage in low-noise Optimice Hepa-filtered ventilated cage racks (Animal Care Systems Inc, Centennial Colorado). All mice were healthy and

immune competent prior to studies the described. Male and female mice were used for experiments. An ear punch or tail snip tissue biopsy was performed to obtain tissue for genotyping when the mice were weaned. Genotyping was performed by Transnetyx using verified protocols. Otherwise, no experimental procedures, drugs, or tests were performed on mice prior to the studies described. Equal numbers of male and female mice aged 10–12 weeks were used for all studies described.

**Generation of experimental models**. The following mice strains were purchased from Jackson laboratories: *C57BL/6J*, *B6.SJL*, *p53$^{-/-}$*, and *Nrp1$^{fl/fl}$* mice. *Sema3A$^{fl/fl}$*

**Fig. 7 BM ECs promote hematopoietic regeneration via secretion of R spondin 2. a** IPA of functions and diseases enriched in BM ECs at 72 h following 500 cGy and treatment with anti-NRP1 relative to irradiated control BM ECs. p values shown for pair-wise comparisons and each replicate is a pool of sorted BM ECs ($n = 5$ mice). **b** Heat map of genes that encode secreted proteins expressed $\geq$ 2-fold in BM ECs from irradiated mice treated with anti-NRP1 compared to BM ECs from irradiated controls. Each replicate is a pool of sorted BM ECs from five mice. **c** R spondin 2 (RSPO2) ELISA of supernatants from BM ECs following 800 cGy irradiation and culture with 10 μg/ml anti-NRP1 or IgG ($n = 3–5$/group; $p = 0.0034$). **d** PB WBC and NEU at day +10 following 500 cGy in mice treated with anti-NRP1, 10 μg anti-R spondin 2, anti-NRP1 + anti-R spondin 2, or IgG every other day × 10 days ($n = 5–7$ mice/group; WBC: $p = 0.015$, NEU: $p = 0.0135$). Dotted lines represent levels in non-IRR controls. **e** (Left) Flow cytometry of BM KSL cells and CD150$^+$CD41/48$^-$KSL LT-HSCs in mice at day +10 following 500 cGy and treatment with anti-NRP1, anti-R spondin 2, anti-NRP1 + anti-R spondin 2, or IgG. Numbers represent percentages in each gate; (Right) % BM KSL and % SLAM KSL cells in each treatment group ($n = 5–7$ mice/group; KSL: $p = 0.007$; SLAM KSL: $p = 0.0123$). **f** Numbers of BM KSL and SLAM HSCs from mice at day +10 following 500 cGy and treatment with anti-NRP1, anti-R spondin 2, anti-NRP1 + anti-R spondin 2, or IgG ($n = 5–7$/group; KSL: $p = 0.0067$; SLAM KSL: $p = 0.0013$). **g** Total BM CFCs and CFU-GEMMs in mice at day +10 following 500 cGy and treatments as shown ($n = 5$/group; CFCs: $p = 0.0002$; GEMM: $p = 0.0026$). **h** Numbers of CFU-GEMMs from BM c-kit$^+$lin$^-$ cells at day +3 following 300 cGy and treatment with 100 ng/ml Wnt3a, 200 ng/ml R spondin 2, or the combination ($n = 3–5$/group; $p < 0.0001$). **c–g** Two-way ANOVA followed by two-sided Holm-Sidak unpaired t-tests; **h** One-way ANOVA with Holm-Sidak's multiple comparison two-sided t-test. *$p < 0.05$, **$p < 0.01$, ***$p < 0.001$, ****$p < 0.0001$. Source Data file provided.

---

mice were developed in the laboratory of Dr. Takeshi Yagi at the National Institute for Physiological Sciences, Japan and were obtained through the RIKEN BioResource Center[56]. *Cdh5-Cre-ERT2* mice were developed in the laboratory of Dr. Ralf Adams[55] and kindly provided by Dr. Shahin Rafii of Weill Cornell University Medicine. *Nrp1*$^{VEGF-}$ mice were kindly provided by Dr. Chenghua Gu, Harvard University.

**TBI and 5FU chemotherapy studies.** For sublethal TBI, adult *C57BL/6J* mice were irradiated with 500 cGy TBI using a Cesium-137 irradiator. Mice were dosed via intravenous injection (IV) every other day for up to 10 days with 10 μg goat anti-NRP1 (R&D Systems, Minneapolis, MN), or goat IgG control (R&D Systems, Minneapolis, MN), beginning at +24 h of post-TBI.

For lethal dose TBI, adult *C57BL/6J* mice were irradiated with 800 cGy TBI, which is lethal at our institution for approximately 50% of *C57BL/6J* mice by day +30 (LD50/30). Mice were dosed IV with either 10 μg goat anti-NRP1 (R&D Systems, Minneapolis, MN) or goat IgG control (R&D Systems, Minneapolis, MN), beginning at +24 h of post-TBI and every other day through day +21.

For evaluation of chemotherapy effects, adult *C57BL/6J* mice were treated with 5 fluorouracil (5FU), 250 mg/kg IV × 1, and then administered 10 μg anti-NRP1 or goat IgG control every other day IV through day +10.

**R spondin 2 inhibition in vivo.** R spondin 2 signaling was blocked in vivo with a commercially available inhibitory rat anti-Mouse R spondin 2 antibody (MAB32661, R&D Systems, Minneapolis, MN). For in vivo inhibition of R spondin 2, mice were treated intraperitoneally (IP) with either 10 μg anti-NRP1 alone, 10 μg anti-R spondin 2 alone, anti-NRP1 plus anti-R spondin 2, or 10 μg rat IgG beginning on day +1 following 500 cGy TBI. Mice were treated every other day for a total of five doses over 10 days.

**BM EC isolation.** BM ECs were isolated following a protocol published by Poulos et al.[6]. Briefly, mice were intravenously labeled with VE-cadherin antibody (25 μg/mouse, BV13-AF647, Biolegend, San Diego, CA). Fifteen minutes following injection, mice were sacrificed, long bones were harvested and crushed with a mortar and pestle in PBS. The crushed bone product was then digested at 37 °C for 8 min in a 1 mg/ml solution of Liberase (Sigma-Aldrich, St. Louis, MO). Following digestion, bones were rinsed in ice-cold complete media (IMDM + 10% FBS), filtered, and then column depleted of lineage committed hematopoietic cells (Miltenyi Biotec, Auburn, CA). EC were then stained with a CD45 antibody (2 μl/10$^6$ cells, Biolegend) and sorted by FACS for the CD45$^-$7AAD$^-$ VE-cad$^+$ cell population. In some studies, additional cell surface markers were used to identify endothelial and perivascular cell subsets. Arteriolar (CD31$^+$Sca-1$^+$) and sinusoidal (CD31$^+$Sca-1$^-$) ECs were distinguished using AF488 CD31 (5 μl/10$^6$ cells, FAB3628G-100, R&D Systems, Minneapolis, MN) and APC-Cy7 Sca-1 (2 μl/10$^6$ cells, 560654, Becton Dickinson (BD) Biosciences, San Jose, CA). Additionally, we assessed endosteal-associated endothelial cells using endomucin (2 μl/10$^6$ cells, bs-5884R-a750 BiosS, Woburn, MA)[8]. For Leptin receptor+(LepR$^+$) stromal cells, CD45$^-$LepR$^+$ cells (1:200, rabbit anti-PE leptin receptor antibody, bs-0961R-PE, BiosS, Woburn, MA) were isolated.

**HSC culture.** BM KSL cells were isolated by FACS of BM Lin$^-$ cells stained with anti Sca-1 APC-Cy7 (2 μl/10$^6$ cells, Biosciences, San Jose, CA), anti c-Kit-PE (2 μl/10$^6$ cells, Biosciences, San Jose, CA), and V450 lineage cocktail (20 μl/10$^6$ cells, BD Biosciences, San Jose, CA), or isotype controls. Sterile cell sorting was conducted on a FACS-Aria cytometer (BD Biosciences, San Jose, CA). BM KSL cells were collected into Iscove's Modified Dulbecco's Medium (IMDM, Life Technologies, Carlsbad, CA) + 10% FBS + 1% penicillin–streptomycin (pen-strep) and irradiated with 300 cGy. Irradiated KSL cells were cultured for 7 days in complete media

containing IMDM, 10% FBS, 1% pen-strep, 125 ng/ml stem cell factor [SCF], 50 ng/ml Flt-3 ligand, and 20 ng/ml Thrombopoietin), with 10 μg/ml goat anti-mouse NRP1 or goat IgG isotype control.

**NRP1 binding assay.** We developed a sandwich NRP1 enzyme linked immuno-sorbent assay (ELISA) to determine the binding capacity of mouse and human SEMA3A to the N-terminal end of mouse NRP1. 6× His-tag microtiter plate wells (ab128573, Abcam, Waltham, MA) were coated with 100 μl/well C-terminal His tagged mouse NRP1 (5994-N1, R&D systems, Minneapolis, MN) or mouse SEMA3A (Sino Biological 50631M07H100, Fisher Scientific, Waltham, MA, positive control) at 50 ng/ml and incubated on a plate shaker (300 rpm) at 4 °C overnight. The next day, plates were washed with 1× wash buffer (ab128573, Abcam, Waltham, MA) and blocked with 200 μl/well of 1% bovine serum albumin (BSA) in PBS containing 0.05% NaN$_3$ at room temperature (RT) for 3 h. Subsequently, wells were washed three times with 1× wash buffer (ab128573, Abcam, Waltham, MA) and mouse SEMA3A Fc Chimera Protein, CF (5926-S3 R&D systems, Minneapolis, MN) or human SEMA3A Fc Chimera Protein (R&D, 1250-S3) was added to the NRP1-coated wells, at increasing concentrations of 0, 1, 2, 4, 8, 16, 32, 128 ng/ml and incubated for 2 h at RT. Plates were washed again three times and 100 μl/well HRP-conjugated SEMA3A detector antibody (bs-10468R-HRP, BiosS, Woburn, MA) was added at 1:250 and incubated for 1 h at RT. Prior to addition of 100 μl/well HRP development solution, plates were washed five times to remove any excess antibody. The blue color development was recorded immediately at 600 nm using a InfiniteM1000 Pro Tecan Plate reader using the Spark Control Software InfiniteM1000 Pro.

**Binding characterization of NRP1 to SEMA3A-Fc by surface plasmon resonance.** NRP1-SEMA3A-Fc binding studies were performed using a Pioneer SensiQ SPR instrument. Mouse NRP1 (5994-N1, R&D systems, Minneapolis, MN), mouse Sema3A/Fc Chimera Protein, CF (5926-S3, R&D systems, Minneapolis, MN) or human Sema3A/Fc Chimera Protein (1250-S3, R&D systems, Minneapolis, MN) were purchased from R&D systems. Lyophilized proteins from R&D Biosystems were resuspended in 20 mM HEPES pH 7.5, 150 mM NaCl and 0.005% Tween-20 that was used as running buffer for all the SPR binding assays. Mouse NRP1 receptor was immobilized on a Polycarboxylate High Capacity (PCH) biosensor chip in 10 mM sodium acetate pH 5.0 to desired RU (2307 RU for mouse NRP1/mouse SEMA3A Fc binding and 1565 RU for mouse NRP1/human SEMA3A Fc assay). SEMA3A-Fc proteins (either human or mouse) were flowed over the immobilized mouse NRP1 to measure binding affinities. OneStep[98] kinetics was used for affinity measurements, wherein the analyte was injected at a single fixed concentration (100 nM mouse SEMA3A Fc or 50 nM human SEMA3A Fc; 75 μl/min flowrate). Binding analysis and affinity calculations were performed using Pioneer QDAT software (Version 3.41).

**Cdk5 inhibition.** Roscovitine is a small molecule that demonstrates selective Cdk5 inhibitory activity at concentrations less than 50 μM[59]. For all in vitro Cdk5 inhibition studies, we used a concentration of 10 μM roscovitine[85] (1332, Tocris Bioscience, Minneapolis, MN).

**CFC assays.** CFC assays [colony-forming unit-granulocyte monocyte (CFU-GM), burst-forming unit-erythroid (BFU-E), and colony-forming unit-mix (CFU-GEMM)] were performed using Stem Cell Technologies Methocult M3434 (STEMCELL Technologies, Vancouver, BC) using the manufacturer's instructions. For BM studies, each CFC dish was a biological replicate loaded with $1 \times 10^4$ BM cells. For measurement of effects of R spondin 2 and/or Wnt3a treatment on CFC recovery following irradiation, CFC dishes were plated with the progeny of $2 \times 10^3$ c-kit$^+$lin$^-$ cells at 72 h following 300 cGy irradiation and culture with media

supplemented with and without 200 ng/ml R spondin 2 alone (6946-RS-025/CF, R&D Systems, Minneapolis, MN), 100 ng/ml Wnt3a alone (1324-WN-002, R&D Systems, Minneapolis, MN) or the combination of R spondin 2 and Wnt3a.

**Competitive repopulation assays**. For competitive repopulation assays, $5 \times 10^5$ BM cells were collected from *C57BL/6J* (CD45.2$^+$) mice at day +10 following 500 cGy TBI and treatment with anti-NRP1 or IgG and transplanted intravenously into lethally irradiated (900 cGy) 10–12 week old congenic B6.SJL (CD45.1$^+$) recipient mice, along with $2 \times 10^5$ CD45.1$^+$ BM competitor cells. Multilineage donor hematopoietic cell engraftment was measured in the PB and BM of recipient mice up to 20 weeks of post-transplant by flow cytometry using the BD FACSDiva Software (v9.1) and analyzed using FlowJo (v10). PB and BM cells were stained with Anti-Mouse CD45.1 Brilliant Violet 605 (2 μl/10$^6$ cells), Mouse Anti-CD45.2 FITC Mouse anti-mouse (2 μl/10$^6$ cells), Rat Anti-Mouse Gr-1 (Ly-6G and Ly-6C) PE (2 μl/10$^6$ cells), Rat Anti-Mouse Mac-1 (CD11b) PE (2 μl/10$^6$ cells), Rat Anti-Mouse CD3 V450 (2 μl/10$^6$ cells), Anti-Mouse CD41 Alexa Fluor 488 (2 μl/10$^6$ cells), Rat anti-Mouse CD150 Alexa Fluor 647 (2 μl/10$^6$ cells).

**Flow cytometric analyses**. BM EC apoptosis and necrosis were measured using BD Biosciences Annexin V-FITC kit or CellEvent Caspase 3/7 Green Flow Cytometry Kit (ThermoFisher Scientific, Waltham, MA). Phospho protein analyses were performed using BD Cytofix/Cytoperm kit and phospho antibodies targeting Cdk5 (2 μl/10$^6$ cells, rabbit anti p-Cdk5 Tyr15, TA325347, OriGene Technologies, Rockville, MD) and p53 (2 μl/10$^6$ cells, anti p-p53 Ser15, 9826S, Cell Signaling Technology, Danvers, MA). Cell cycle analysis was performed using BD Cytofix/Cytoperm kit and the BD FITC Mouse anti-Ki-67 Set with 7AAD (2 μl/10$^6$ and 5 μl/10$^6$ cells, respectively). Cell surface expression of LGR5 was measured on HSC and progenitor cell populations via staining with AF488-conjugated anti-mouse LGR5 (R&D systems, Minneapolis, MN). Detailed information about all reagents is included in Supplemental Table 1.

**BM EC permeability**. For analysis of BM EC permeability, we used an Evans Blue Dye (EBD) extravasation assay[13]. Briefly, *C57BL/6J* mice were irradiated with 5Gy TBI and subsequently treated with SEMA3A or anti-NRP1. At +24 h post-TBI, mice were injected IV with 200 μl of a 0.5% solution of EBD. Fifteen minutes later, BM supernatants were collected and EBD concentration was determined by measuring absorbance on a spectrophotometer at 610 nm.

**Histology**. Mice were intravenously labeled with anti-VE-cadherin antibody (25 ug/mouse, Biolegend;BV13-AF647). Fifteen minutes following injection, mice were sacrificed and femurs were harvested. For thick histology sections, femurs were prepared by fixing femurs for four hours in 4% paraformaldehyde on ice and then decalcifying for 24 h in 0.5 M EDTA, pH 7.4–76 at 4 °C[99]. Femurs were then cryoprotected in cryoprotectant solution (200 g sucrose, 20 g PVP, 700 mL PBS) and embedded in embedding solution (8 g gelatin, 2 g PVP, 20 g sucrose, 80 mL PBS). For thin histology sections, femurs were fixed overnight at 4 °C in 4% paraformaldehyde followed by 2 days in 10% EDTA (pH 7.4) at 4 °C to decalcify bones. Bones were then rinsed in PBS and transferred to a 20% sucrose solution for 2 h at room temperature. Bones were then rapidly embedded in OCT medium on dry ice. Femurs were sectioned at 8 μm and mounted using ProLong Gold Antifade Mountant with DAPI (ThermoFisher). For thin sections, images were acquired at ×10 and ×20 magnification on a Zeiss Axio Imager M2 using the Zen 2 (blue) edition software. For thick sectioning, femurs were sectioned at 100 μm[99]. Sections were immunostained by rehydrating the section in PBS, permeabilizing in 0.5% Triton-X, and blocking with 5% bovine serum albumin. Sections were then stained with antibodies to CD31 (1:40, Biolegend), VE-Cadherin 1:40, Biolegend), SEMA3A (1:200, [EPR19367] (ab199475, Abcam) and NRP1 (1:200, [EPR3113] ab81321, Abcam) and and DAPI (Supplementary Table 1), mounted and imaged. Thick sections were imaged using a Leica SP8 (SMI8-CS) confocal microscopy system using the Leica LAS X Version 5.0.2 with a 20× objective (HC PL APO CS2 20×/0.75 IMM) and a white light laser for excitation and two HyD photon counting detectors. Z-Stacks were acquired in sequential mode using a step size of 5 μm with a scan speed of 8000 Hz. Images were deconvolved using the Leica lightning software with setting for Fluoromount mounting media.

**Quantitative RT-PCR**. RNA was isolated using the Qiagen RNeasy micro kit. RNA was reverse transcribed into cDNA using the High Capacity cDNA reverse transcription kit (Applied Biosystems). Real time PCR analysis was performed using Taqman Gene Expression assays on an Applied Biosytems QuantStudio 6 Real Time PCR Machine with the QuantStudio Analysis Software (ver 1.7.1) with the following probe sets: Sema3A Mm00436469_m1, Nrp1 Mm00435379_m1, Nrp2 Mm00803099_m1, Sema3f Mm00441325_m1, Sema7a Mm00441361_m1, Sema6a Mm00444441_m1. Gene expression was normalized to *Gapdh* level for each sample using the $2^{-\Delta\Delta CT}$ method.

**SEMA3A ELISA**. For ELISA of BM supernatants, BM from a single femur was flushed with an insulin syringe into 300 μl PBS. BM cells were pelleted by

centrifugation and the undiluted supernatant was assayed by ELISA using a sandwich ELISA kit purchased from Biomatik (EKU07258). For cell culture supernatants, multiple wells of a 24-well plate were pooled and then concentrated 10-fold using an Amicon Ultra 10K centrifugal filter unit prior to ELISA.

**R spondin 2 ELISA**. For ELISA of cultured BM EC supernatants following irradiation and treatment with anti-NRP1, primary murine BM ECs were cultured in 24-well cell culture dishes to confluence in EGM-2 endothelial cell growth medium supplemented with FBS, Hydrocortisone, hFGF-B, VEGF, R3-1GF, ascorbic acid, hEGF, GA-1000, and Heparin, per the manufacturer's recommendation (Lonza, Basel, Switzerland). Confluent EC cultures were irradiated with 800 cGy and treated with either 10 μg/ml goat-IgG or goat anti-mouse NRP1 (R&D Systems). Cell culture supernatants were collected at day +7 of post-irradiation and treatment. Supernatants were filtered to remove cellular debris and assayed undiluted for RSPO2 protein using a mouse RSPO2 sandwich ELISA kit purchased from Lifespan Biosciences (LS-F39113-1).

**BM vascular perfusion**. For perfusion studies, 100 μg of Lycopersicon Esculentum (Tomato) Lectin (LEL, TL), DyLight® 594 (Vector Laboratories) was perfused into mice via tail vein injection. Mice were sacrificed 15 min following injection and femurs were prepared according to Kusumbe et al.[99] and visualized using a Leica SP8 microscope system for lectin localization within the bone marrow. The mean fluorescence intensity of lectin staining found within the VE-cadherin+ vascular area was quantified using ImageJ.

**Quantification of vascular area**. Using Fiji/ImageJ (Version 2.3.0/1.53f, NIH, Bethesda, MD), image thresholds were developed using the same parameters for all conditions. The percent vascular area was quantified by calculating the percentage of the field of view with VE-cadherin vascular staining.

**RNA sequence analysis**. RNA from sorted BM ECs was isolated using the Qiagen RNeasy micro kit. The UCLA Technology Center for Genomics & Bioinformatics performed the RNA-seq analysis using established methods for library construction and sequencing. Libraries for RNA-Seq were constructed with Clontech Kit to generate strand-specific RNA-seq libraries. The workflow consisted of poly(A) RNA selection, RNA fragmentation and double-stranded cDNA generation using a mixture of random and oligo(dT) priming, followed by end repair to generate blunt ends, adapter ligation, strand selection, and PCR amplification to produce the final libraries. Amplified libraries were quantified by Qubit dsDNA HS (High Sensitivity) Assay Kit, and quality-checked by the Agilent 4200 TapeStation System. Different index adapters were used for multiplexing samples in one sequencing lane. Sequencing was performed with NextSeq500 High Output sequencer to produce 75 base-pair single-end reads (1 × 75 bp). Raw data were deposited in the GEO repository under the accession number GSE149776.

The Partek Flow analysis and Ingenuity Pathway Analysis (IPA) were used for bioinformatics methods and data analysis. Reads per gene were quantified using STAR-2.7.2a [3] and mm10 (Ensembl GRCm38.97). After obtaining gene counts, the counts were normalized by CPM. The differential gene expressions were examined by GSA. For all results of differential gene expression analysis, statistical filters were applied: $p < 0.05$, FDR < 0.05, and fold change |FC|>2. Within the differentially expressed gene set, QIAGEN's Ingenuity® Pathway Analysis (IPA®, QIAGEN Redwood City, www.qiagen.com/ingenuity) was applied to identify enriched canonical pathways, diseases and biological functions. Secreted differentially expressed genes were identified by comparing our differentially expressed gene set with *MetazSecKB*: the human and animal secretome and subcellular proteome knowledgebase of secreted proteins[62].

**Statistics and reproducibility**. Values are reported as means ± SEM unless stated otherwise. All comparisons performed were two-tailed Student's *t*-test, unless otherwise indicated in the Figure Legends. GraphPad Prism 6.0 was used for all statistical analyses. All data were checked for normal distribution and similar variance between groups. Data were derived from multiple independent experiments from distinct mice or cell culture plates. Sample sizes for in vitro studies were chosen based on observed effect sizes and standard errors from prior studies. For all animal studies, a power test was used to determine the sample size needed to observe >30% differences between groups with 0.8 power using a two-tailed student's *t*-test and *p* value of 0.05. All animal studies were performed using age-matched animals. Animal studies were performed without blinding of the investigator. No data were excluded from the analyses. The experiments were not randomized.

**Reporting summary**. Further information on research design is available in the Nature Research Reporting Summary linked to this article.

## Data availability

The raw data files of RNA sequencing data have been deposited and are openly availability at the Gene Expression Omnibus GEO data repository under accession code

(GSE149776. The MetazSecKB database is available at [http://proteomics.ysu.edu/secretomes/animal/]. All other data supporting the findings of this study are openly available within the Source Data File included in this manuscript. Source data are provided with this paper.

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

## Acknowledgments

The authors wish to thank Lisa T. Uechi and Michael J. Mashock from the UCLA Technology Center for Genomics and Bioinformatics for their assistance with the RNA sequence analysis. We thank Karen Lyons and William McBride for facility access, Edo Israely for FACS assistance, the Advanced Light Microscopy/Spectroscopy Laboratory and the Leica Microsystems Center of Excelence at the California NanoSystems Institute at UCLA, and Dr. Laurent Bentolila and Dr. Matthew Shibler for assistance with imaging. This work was supported, in part, by NHLBI grant, HL-086998 (to J.P.C.), NIAID grant AI-067769 (to J.P.C.), California Institute for Regenerative Medicine Leadership Award LA1-08014 (to J.P.C.), NIH K01 1K01DK126989-01A1 (C.M.T.), Damon Runyon Cancer Foundation DRG-2327-18 (C.M.T.), the Burroughs Wellcome Fund PDEP #1018686 (C.M.T.), the UC President's Postdoctoral Fellowship (C.M.T.), NIH K08 1K08HL138305 (V.Y.C.), NIAID grant AI-138331 (to H.A.H.) and the Tower Cancer Research Foundation Career Development Grant (to M.R.).

## Author contributions

J.P.C. and C.M.T. designed and J.P.C. directed the study. C.M.T., A.P., T.F., M.R., V.Y.C., Y.Z., N.S., L.S., M.L., M.M.K., O.T., P.K.L., J.P.S. and H.A.H. performed the experiments reported in the study. A.C. and R.M. performed biophysical studies. C.M.T. and J.P.C. wrote the manuscript with input from the other authors.

## Competing interests

The authors declare no competing interests.
