## [Peer Review File · Nature Communications]

Neuropilin 1 regulates bone marrow vascular regeneration and hematopoietic reconstitutionREVIEWER COMMENTS

Reviewer #1 (Remarks to the Author):

General Comments:

This is a strong paper that presents new information, not previously known or realized, that implicates SEMA3A-NRP1 signaling and R spondin 2 in the damage inflicted on bone marrow endothelial cells by radiation and 5FU, that then greatly compromises hematopoietic recovery. The investigators use multiple means (including antibodies, antagonists, relevant genetic mouse models, etc.) to make the case for the involvement of these signals in the damage inflicted. Most importantly they use this information to show how it is possible to counter the damage and accelerate hematopoietic cell recovery. My comments below are minor.

Minor Comments:

1) In a number of places the investigators report data as a percent or frequency, when the information would be more meaningful if presented as absolute numbers (e.g. per femur). This is especially important when assessing in vivo effects. Examples of this are for Fig. 3c, 4g, f, 5h, 7 f-h, etc.

2) Fig. 3d. Do you have any secondary transplant data for this experiment. If so, please add. If not, that is OK.

3) Please make sure that the keys, where necessary, are shown directly on the Figure itself so that the reader does not have to look at the legends to figure out what the different groups are (e.g. Fig. 3a).

--

Reviewer #2 (Remarks to the Author):

The manuscript by Himburg et al describes an interesting set of experiments in which the relevance of Semaphorin-3A and its receptor Neuropilin for niche-dependent HSC regeneration after irradiation is described. Recently, several papers have appeared in which molecules initially thought to only be relevant for neurogenesis have been shown to also affect HSC biology. The connection between the neural system and the hematopoietic system is very poorly understood, yet may be of significant biological relevance. In the present paper data are provided that suggest that ligand-receptor interactions previously unrecognized in the regulation of the hematopoietic system exist, and are relevant. Therefore, the relevance and novelty of this study is high. Overall, the experiments are carefully carried out and are well described. Nevertheless, beyond several issues detailed below, I did find the data that mechanistically link Sema3A-Nrp1 with Lgr5 and Rsp2 not convincing. These data need more solid validation. Below I provide my comments:

1. The authors report the differential expression of several novel receptor-ligands by endothelial bone marrow cells, most notably Sema3a, Sema3f and Nrp1, in steady state, and upon irradiation. Recently several studies have performed extensive single cell profiling of various bone marrow niche cell fractions, and it would be of interest to assess to what extent the data reported in the current study agrees -or not- with these single cell papers. I performed some analysis using the online niche tool (<https://nicheview.shiny.embl.de>). Differential expression of Nrp1 in distinct cell subsets is very clear, but Sema3a and Sema3f appear to be expressed at below-threshold values. How do the authors reconcile their data with these findings?

2. The Y-axis scale in Figure 1A is unclear to me. The legend reads that 'data are normalized to non-irradiated Sema3a expression'. How can data for Sema3f or Cdh5 be compared and normalized to

qPCR data for *Sema3a*? These measurements were done with different primers and PCR conditions, so I cannot understand how it is possible to suggest that *Sema3f* values in normal cells are ~50% of that of *Sema3a*, as the black circles in panel 1A suggest. RNA-Seq experiments would allow to compare read counts between genes, but using qPCR is it very difficult to compare genes.

3. It is highly surprising that the authors identify the expression of *Lgr5* on a very significant population of primitive HSCs, and document upregulation of *Lgr5* upon irradiation. There are many gene expression studies in which HSCs have been transcriptionally profiled (several of them available at Bloodspot (<http://servers.binf.ku.dk/bloodspot/>) and consistently, there is no record of *Lgr5* expression in HSCs. Also, the *Lgr5* reporter mouse made by the Clevers lab does not show any evidence of its expression in HSCs. The involvement of *Lgr5* and *Rsp2* in the hematopoietic system is therefore surprising; many people have looked but this has led to nothing. The authors do not show primary FACS plots in figure 7, so it is unclear to what extent these data are solid. FACS plots need to be shown, and qPCR data on CD34-KLS cells pre- and post-radiation need to confirm this strong upregulation.

4. The effects of antiNRP1 and antiRsp2 treatment as depicted in Figure 7e and f are very subtle, and taken at one specific time point at which the bone marrow is rapidly/exponentially recovering from the irradiation. At these timepoints (day 7 /day 10) the bone marrow cellularity is at its nadir. As depicted (i.e. without showing values for unirradiated control mice) the effects seem apparent, but if the authors would include values for unirradiated control mice it would become clear that although statistically significant, biological relevance of these very small changes are questionable.

5. In the discussion the authors claim that *Rsp2* was essential for recovery of the HSC pool, but clearly, that conclusion cannot be drawn. HSCs recover just fine upon inhibiting of *Rsp2*.

6. Are there any hematopoietic data available for the mice that are shown in panel 3f? These survival curves are clear, but it would be informative to assess whether indeed peripheral blood cell recovery coincides with survival.

--

Reviewer #3 (Remarks to the Author):

In their manuscript, Himburg and colleagues investigate the molecular mechanisms by which bone marrow (BM) endothelial cells (ECs) regulate hematopoietic regeneration upon myeloablation caused by total body irradiation (TBI) or chemotherapy. Based on their experimental data, the Authors propose a model in which, in response to myeloablation, BM ECs secrete Semaphorin 3A (SEMA3A) that inhibits BM vascular regeneration by acting via NRP1 on ECs themselves. SEMA3A/NRP1 signaling would promote EC apoptosis via Cdk5-mediated phosphorylation of p53 and increased transcription of p53-upregulated modulator of apoptosis (PUMA). Upon myeloablation, NRP1 blockade also promotes the production of Wnt signal enhancer R spondin 2 (*Rspo2*) in BM ECs and the expression of *Rspo2* receptor LGR5 in hematopoietic stem/progenitor cells (HSPCs). Combination of *Rspo2* and *Wnt3a* promotes hematopoietic regeneration upon TBI. This manuscript could provide interesting findings on the role of NRP1 in the control of BM hematopoiesis, however several important issues significantly hamper the strength of this work.

Major issues

1. The expression of SEMA3A in BM ECs and the ability of SEMA3A to inhibit BM angiogenesis were already reported by Vacca et al. (*Blood*, 2006) and Lavi et al. (*Carcinogenesis*, 2018), respectively. These published articles should be cited, mentioned and discussed where appropriate throughout the manuscript. Considering the existing previous work, the novelty of findings reported by Himburg et al.

is lessened, yet the NRP1 link to Rspo2/LRG5 signaling is novel and previously unexplored.

2. For in vivo studies, two key reagents were employed: recombinant murine SEMA3A (R&D Systems) and goat anti-NRP1 (R&D Systems).

A. High affinity SEMA3A/NRP1 interaction depends on the binding of C-terminal SEMA3A C-end rule (C-endR) R/KXXR/K peptide(s) (Teesalu et al., PNAS 2009;), exposed by furin cleavage (Adams et al., EMBO J. 1997), to the b1 domain of NRP1 (Guo & Vander Kooi, J. Biol. Chem., 2015 and summarized in Fig. 2 of Kumanogoh & Kikutani, Nat Rev. Immunol. 2013). In the C-terminal portion of SEMA3A different furin cleavage sites exist that give rise to different furin-processed SEMA3A variants endowed with different affinity for the b1 domain of NRP1 (Guo et al., Biochemistry, 2013). A crucial aspect is that unfortunately, differently from the human version produced by the same company, the commercial recombinant mouse Sema3A-Fc Chimera from R&D Systems (Catalog Number: 5926-S3) stops at Lys 747 and lacks the last 25 amino acids that contain the two highest affinity binding sites (RAPR & RNRR; see Guo et al., Biochemistry, 2013) for the b1 domain of NRP1. Therefore, data obtained by injecting mouse Sema3A-Fc Chimera from R&D Systems are extremely difficult to be interpreted and it is difficult to reconnect the observed effects with high affinity SEMA3A binding to NRP1. To support the model they are proposing, Authors must employ a purified recombinant SEMA3A that they can prove to bind NRP1 with high affinity.

B. The fact that, as reported in the data sheet, affinity-purified polyclonal goat IgGs anti-mouse/rat NRP1 from R&D Systems (Catalog Number: 5926-S3) displace the binding of both human VEGF-A 165 and human SEMA3A-Fc chimera to immobilized recombinant rat NRP1-Fc chimera clearly suggests that this polyclonal goat IgGs inhibit the binding of different ligands to NRP1 b1 domain. Since the b1 domain of NRP1 is capable of binding all ligands containing C-endR peptides and not only (e.g. He et al., Nature 2015), it is formally difficult to attribute the in vivo effect of 5926-S3 goat anti-mouse/rat NRP1 IgGs to an impaired SEMA3A binding only.

3. In general, the phenotypes of BM ECs and blood vessels are quite poorly analyzed and must be more carefully characterized and quantified. Authors are limiting their analyses to single sections of not otherwise defined portion(s) of the femoral BM cavity, staining only VE-cadherin, which is not an ideal marker due to its concentration at cell-to-cell contacts. BM ECs of blood vessels must be stained with DAPI, anti-endomucin, anti-CD31, and α -smooth muscle actin, and then analyzed with high resolution confocal microscopy and 3D reconstruction (e.g. see Kusumbe et al., Nat. Protocols, 2015).

4. Are treatments (TBI, SEMA3A, anti-NRP1) impacting on blood vessel perfusion (e.g. use fluorescently labeled lectins) and function (e.g. monitor permeability with fluorescent dextran)?

5. How can authors count VE-cad+ BM ECs without imaging cell nuclei?

Minor issues

On page 10, second para, referring to references 57 and 58, Authors state that SEMA3A signals via Plexin A2. It is now known that both in neurons (Yaron et al., Neuron, 2005) and ECs (Kigel et al., Blood, 2011) SEMA3A mainly signals via Plexin A4 and not Plexin A2, which is instead exploited by SEMA3B (Sabag et al., J. Cell Sci., 2014).

--

Reviewer #4 (Remarks to the Author):

The manuscript "Semaphorin 3A controls bone marrow vascular regeneration and R spondin 2 – mediated hematopoietic reconstitution" submitted by Himburg and colleagues studies the role of bone marrow (BM) endothelial cells (EC) in vascular and hematopoietic stem cell (HSC) regeneration after myeloablation using irradiation or chemotherapy. The authors identified that BM ECs secrete semaphorin 3A (SEMA3A) in response to myeloablation. SEMA3A binds to its receptor Neuropilin (NRP1) and induces EC apoptosis via activation of cyclin dependent kinase 5 and p53, which represses vascular regeneration. The deletion or inhibition of NRP1 reduces EC apoptosis and resulted in an accelerated BM vascular regeneration as well as HSC regeneration. Using RNA sequencing the authors identified R spondin 2 as being highly secreted in irradiated ECs upon NRP1 inhibition as well as an increased expression of the R spondin 2 receptor LGR5 in HSCs. The inhibition of R spondin 2 blocked HSC regeneration in response to NRP1 inhibition confirming the link between inhibition of SEMA3A-NRP1 signaling and secretion of R spondin 2 to promote BM EC and HSC regeneration after myeloablation.

This study aims to better understand the molecular mechanism and cellular interactions in the BM niche that mediate vascular and HSC regeneration after myeloablation. This knowledge is highly relevant to develop targeted therapies to improve vascular and HSC regeneration after irradiation or chemotherapy to reduce the time of myelosuppression in the patient. This work will be relevant to many readers in the hematopoietic field.

The presented study is well designed and easy to follow as a reader. The applied methodology represents state of the art techniques. The method section provides in a large extend enough details to reproduce the work. The data seems to be robust and supports most conclusions of the authors. However, some aspects need clarification and some additional data to support the author's findings.

Please find these points of criticism and suggestions in the following list:

1. In Figure 1a, the expression of Cdh5 was measured in VE-Cad+ BM EC. It is unclear to the reader why this was done as it is not mentioned in the main text. Moreover, it is not clear why the expression is so low although presumably sorted VE-Cad+ BM ECs were used in the qPCR? Please comment on this aspect and mention at least once that VE-Cad and Cdh5 are the same gene. Especially for readers who are unfamiliar with ECs, this information will be helpful to better understand the applied mouse models later.
2. In Figure 1b, what is the frequency of sBMECs and aBMECs? A FACS dot plot showing the gating strategy would be helpful in the extended materials. How many cells were sorted and used for qPCR analysis? Please indicate this number in the method section.
3. Did the authors perform any statistics on Figure 1d? In case there is no statistical significance, please still indicate the applied statistical test and that the result is not significant.
4. In Figure 1h, it is not clear to the reader why endomucin+ ECs were included in the analysis of NRP1+ cells. Please comment on this aspect in the main text.
5. Additionally, in Figure 1h, please mention in the figure legend what the black line in the histogram refers to. Moreover, as there is a shift in the intensity of NRP1 expression an additional graph showing the MFI of NRP1 would be of great value to demonstrate increased expression of the receptor.
6. In Figure 2, the authors claim that the administration of anti-NRP1 accelerates BM vascular regeneration. However, there is no data presented showing that the BM niche is destroyed after irradiation and that it is regenerated more quickly. The presented data only shows one time point (day 7 after irradiation) at which the regeneration presumably already had occurred. It is necessary to show that the BM vasculature was damaged at earlier time points. The authors could perform a time course experiment showing the regeneration of BM vessels at different time points in the different conditions (IgG, SEMA3A and anti-NRP1). This experiment would be helpful to proof that an

accelerated regeneration has occurred. Otherwise one might wonder whether anti-NRP1 administration has a protective effect.

7. Is it possible to score or quantify the damage of VE-Cadh+ BM vessels that are depicted in Figure 2a and other similar figures? Maybe similar as performed in Extended Figure 3b. A score or quantification would be helpful to assess the extend of damage and regeneration after irradiation and would strengthen the findings much more than just representative images of the BM vasculature.
8. In Figure 2e, the authors write in the main text and illustrate in the figure that mice were irradiated with 700 cGy prior to the BM EC permeability assay with Evans Blue Dye. Contrarily, in the method section it is written that 500 cGy were used. Please clarify this aspect. Why were 700 cGy used although in all previous experiments 500 cGy were used? How permeable are BM ECs after 500 cGy irradiation?
9. Please mention in the figure legend of Figure 3b and 3d what the dotted line refers to.
10. Please indicate in the figure legend of Figure 4a to what the qPCR results were normalized to.
11. Please mention in the figure legend of Figure 4e what the dotted line refers to.
12. In Figure 4 and 5, to further strengthen the finding that NRP1 inhibition improves HSC regeneration it would be helpful to include a CFU Assay to measure GEMM colonies and quantification of LT-HSCs. Ultimately a competitive bone marrow transplantation as done in Figure 3a would be of great value.
13. At baseline, floxed Sema3a and Nrp1 mice have different CD31+ cell counts (0.2% Figure 5e versus 0.4% Extended Data Figure 3c). Assuming that the floxed animals are on the same background how is this discrepancy explained? Is there any biological mechanism that could contribute to this observation? Or is this interexperimental variation? How sensitive is then the measurement of VE-Cadh+ cells?
14. Please indicate in the figure legend of Figure 5a to what the qPCR results were normalized to.
15. In Figure 6d, in addition to Caspase3/7 activation it would be helpful to also show the frequency of Annexin V+ cells to prove presence of apoptotic cells.
16. Additionally, a one-way ANOVA was used to analyze the data in Figure 6d. The authors might use instead a two-way ANOVA here as two parameters were used.
17. In Extended Data Figure 5a the expression levels of different p53 induced genes are shown. Noxa is mentioned in the main text as such a gene but the data is not presented. Is there a reason for it? Could Noxa expression being added to the figure?
18. In contrast, Puma expression is shown twice in Figure 6e and Extended Data Figure 5a. The data does not need to be shown again in the Extended Data.
19. In Figure 7d also Lin- and LKS- cells were analyzed but the data is not mentioned in the main text. Please mention all your results in the main text.
20. For the cell cycle analysis in Extended Data Figure 1c, a representative FACS dot blot showing the gating strategy would be helpful for the reader to better understand the data collection.
21. Similar to comment #3, did the authors perform any statistics on Extended Data Figure 1f, 2b-e and 3b, d-f? In case there is no statistical significance, please still indicate the applied statistical test and that the result is not significant.
22. Please indicate in the figure legend of Extended Data Figure 5a to what the qPCR results were normalized to.

Response to Reviewers

We very much appreciated the thorough review that our original manuscript received and we have extensively revised the manuscript in response to the Reviewers' comments. The Reviewers' comments are shown below in italics, along with our point-by-point responses:

Reviewer 1

General Comments:

This is a strong paper that presents new information, not previously known or realized, that implicates SEMA3A-NRP1 signaling and R spondin 2 in the damage inflicted on bone marrow endothelial cells by radiation and 5FU, that then greatly compromises hematopoietic recovery. The investigators use multiple means (including antibodies, antagonists, relevant genetic mouse models, etc.) to make the case for the involvement of these signals in the damage inflicted. Most importantly they use this information to show how it is possible to counter the damage and accelerate hematopoietic cell recovery.

We appreciate the Reviewer's positive comments on our manuscript.

My comments below are minor.

Minor Comments:

1) In a number of places the investigators report data as a percent or frequency, when the information would be more meaningful if presented as absolute numbers (e.g. per femur). This is especially important when assessing in vivo effects. Examples of this are for Fig. 3c, 4g, f, 5h, 7 f-h, etc.

We thank the Reviewer for this suggestion and have edited Figures 3c, 4g, 5h and 7f to show cell numbers rather than percentages.

2) Fig. 3d. Do you have any secondary transplant data for this experiment. If so, please add. If not, that is OK.

We did perform competitive secondary transplantations of BM cells collected at 20 weeks post-transplant from primary recipient mice, comparing the irradiated/anti-NRP1 treatment group versus irradiated/control treatment groups. We did not detect differences in secondary donor cell engraftment through 20 weeks post-transplant between the treatment groups. These results are shown in revised Supplementary Fig. 2d. The lack of detectable difference in the secondary transplant studies may have been due to our performance of competitive secondary transplantation assays, rather than non-competitive secondary transplant assays. Nonetheless, our results suggest that the effect of anti-NRP1 treatment on BM vascular regeneration may have predominantly affected the regeneration of BM HSCs capable of primary competitive repopulation, rather than longer term – HSCs with serial repopulating capacity.

3) Please make sure that the keys, where necessary, are shown directly on the Figure itself so that the reader does not have to look at the legends to figure out what the different groups are (e.g. Fig. 3a).

We have edited all of the Figures so that legends describing the different groups are included in the Figures themselves.

Reviewer 2

*The manuscript by Himburg et al describes an interesting set of experiments in which the relevance of Semaphorin-3A and its receptor Neuropilin for niche-dependent HSC regeneration after irradiation is described. Recently, several papers have appeared in which molecules initially thought to only be relevant for neurogenesis have been shown to also affect HSC biology. The connection between the neural system and the hematopoietic system is very poorly understood, yet may be of significant biological relevance. In the present paper data are provided that suggest that ligand-receptor interactions previously unrecognized in the regulation of the hematopoietic system exist, and are relevant. Therefore, the relevance and novelty of this study is high. Overall, the experiments are carefully carried out and are well described. Nevertheless, beyond several issues detailed below, I did find the data that mechanistically link *Sema3A-Nrp1* with *Lgr5* and *Rsp2* not convincing. These data need more solid validation. Below I provide my comments:*

*1. The authors report the differential expression of several novel receptor-ligands by endothelial bone marrow cells, most notably *Sema3a*, *Sema3f* and *Nrp1*, in steady state, and upon irradiation. Recently several studies have performed extensive single cell profiling of various bone marrow niche cell fractions, and it would be of interest to assess to what extent the data reported in the current study agrees -or not- with these single cell papers. I performed some analysis using the online niche tool (<https://nicheview.shiny.embl.de>). Differential expression of *Nrp1* in distinct cell subsets is very clear, but *Sema3a* and *Sema3f* appear to be expressed at below-threshold values. How do the authors reconcile their data with these findings?*

We appreciate the Reviewer's feedback and suggestions to evaluate our BM EC gene expression results in the context of publicly available datasets of single bone marrow niche cell gene expression analyses. We also find it somewhat surprising that *Sema3a* is not demonstrated to have increased expression in the endomucin⁺ sinusoidal ECs or the Ly6a (Sca-1)⁺ arteriolar ECs in the <https://nicheview.shiny.embl.de> dataset (Baccin et al. *Nat Cell Biol* 22;38-48, 2020). However, we also examined single cell gene expression analysis of BM ECs performed by the laboratory of Dr. Ianis Aifantis, New York University, and published recently (Tikhonova A et al. *Nature* 69;222-228, 2019). Utilizing the NYUMC publicly available tool, <https://compbio.nyumc.org/niche/>, we found that *Sema3a* and *Nrp1* are strongly expressed by Stab2⁺ BM ECs (termed the "V2" EC cluster) and to a lesser extent in the Ly6a⁺ BM ECs (V1 cluster). Further, *Sema3a* expression was increased in Ly6a⁺ ECs following 5FU chemotherapy compared to homeostasis in the NYUMC dataset. We further assessed our findings by examining *Sema3a* and *Nrp1* expression in single BM EC populations by utilizing the online BM niche analysis tool produced by the Scadden Laboratory and MIT, https://singlecell.broadinstitute.org/single_cell/study/SCP361/mouse-bone-marrow-stroma-in-homeostasis#study-visualize (Baryawno et al. *Cell* 177;1915-1932, 2019). These authors identified populations of BM ECs demonstrating a spectrum of expression of EC markers, Pecam1, Cdh5, Cd34, Kdr and Endomucin and segregated single BM EC populations based on high expression of Vegfr3 and low expression of Ly6a (Vegfr3+Ly6a-) as sinusoidal ECs and Vegfr3-Ly6a+ ECs as arteriolar ECs. Utilizing this tool, we found that *Sema3a* was highly enriched in expression in the Vegfr3+Ly6a- sinusoidal BM ECs, with much less expression noted in Ly6a+ arteriolar ECs. Based on our analysis of these publicly available datasets from the Aifantis and Scadden laboratories, this would suggest that a population of Ly6a- sinusoidal BM ECs express *Sema3a* and *Nrp1*, which is consistent with our results shown in revised Fig. 1c and 1e. The differences in expression of *Sema3a* in sinusoidal BM ECs between these

datasets may be related to the distinct markers that were utilized to define sinusoidal BM ECs in each study. We have added comment to the Discussion to address our findings relative to that of these 3 important studies of single cell BM EC gene expression analyses.

*2. The Y-axis scale in Figure 1A is unclear to me. The legend reads that 'data are normalized to non-irradiated *Sema3a* expression'. How can data for *Sema3f* or *Cdh5* be compared and normalized to qPCR data for *Sema3a*? These measurements were done with different primers and PCR conditions, so I cannot understand how it is possible to suggest that *Sema3f* values in normal cells are ~50% of that of *Sema3a*, as the black circles in panel 1A suggest. RNA-Seq experiments would allow to compare read counts between genes, but using qPCR is it very difficult to compare genes.*

We appreciate the Reviewer's comment and agree with this concern. In response, we have revised Fig. 1a to show a heat map of RNAseq analysis of multiple semaphorin genes within CD45⁺VEcad⁺ BM ECs in homeostasis and following 500 cGy TBI. Via this analysis, we detected *Sema3a* gene expression at baseline in BM ECs and observed increased expression following irradiation. We have also revised Fig. 1b to show qRTPCR analysis of select semaphorin genes and neuropilin 1 and 2 in BM ECs from non-irradiated mice and at 24 hours following 500 cGy TBI. In revised Fig. 1b, normalization of expression of each gene after TBI is made to the level of expression of the same gene in non-irradiated BM ECs.

*3. It is highly surprising that the authors identify the expression of *Lgr5* on a very significant population of primitive HSCs, and document upregulation of *Lgr5* upon irradiation. There are many gene expression studies in which HSCs have been transcriptionally profiled (several of them available at Bloodspot (<http://servers.binf.ku.dk/bloodspot/>) and consistently, there is no record of *Lgr5* expression in HSCs. Also, the *Lgr5* reporter mouse made by the Clevers lab does not show any evidence of expression in HSCs. The involvement of *Lgr5* and *Rsp2* in the hematopoietic system is therefore surprising; many people have looked but this has led to nothing. The authors do not show primary FACS plots in figure 7, so it is unclear to what extent these data are solid. FACS plots need to be shown, and qPCR data on CD34-KLS cells pre- and post-radiation need to confirm this strong upregulation.*

We appreciate the Reviewer's comments regarding *Lgr5* expression on HSCs and recognize that prior analyses, particularly that of the Clevers' laboratory utilizing the *Lgr5* reporter mice, have not suggested that adult HSCs express *Lgr5* or a functional role for *Lgr5* in HSCs. We thank the Reviewer for providing the link to the *Bloodspot* datasets, which also suggests very low or absent expression of *Lgr5* in human and murine HSCs, respectively. We therefore repeated our flow cytometric analysis of BM 34⁺KSL HSCs for surface expression of LGR5 protein at baseline and following 500 cGy irradiation of mice. We also utilized 2 different anti-LGR5 antibody clones (OT12A2 and 803420 from R&D Systems) in order to validate our flow cytometric results. We also performed gene expression for *Lgr5*, as well as *Lgr4* and *Lgr6* in BM 34⁺KSL cells before and after TBI. Since R spondins have been shown to mediate Wnt amplifying effects independently of LGRs (Lebensohn, A., Rohatgi, eLIFE 7, e33126, 2018), and through other receptors such as Frizzled 8 and LRP6 (Nam et al. *J Biol Chem* 281;13247-13257, 2006; Kim et al. *Science* 309;1256-1259, 2005; Wei et al. *J Biol Chem* 282;15903-15911, 2007), we also evaluated gene expression of these candidate receptors in HSCs before and after TBI. In our repeat flow cytometric analyses, we set conservative gating to exclude any non-specific staining within the LGR5⁺ gate. In our repeat studies, we did not confirm LGR5 surface expression or an increase in LGR5 surface expression on BM 34⁺KSL cells from mice irradiated with 500 cGy TBI (Supplementary Fig. 7b-d). We detected low baseline gene expression of *Lgr5*, *Lgr4*, *Lgr6*, *Fzd8* and *Lrp6* in BM 34⁺KSL cells and no increase in expression of these genes following 500 cGy TBI (Supplementary Fig. 7e). Taken together, these results suggest that TBI does not significantly induce transcription or surface expression of *Lgr5* on BM

HSCs. We have edited our Discussion to remove speculation as to the effect of TBI on Lgr5 expression by BM HSCs. Since R spondins can mediate Wnt amplifying effects in the absence of LGRs, we will explore in future studies the specific receptor responsible for mediating R spondin 2 effects on hematopoietic regeneration and whether such effects occur directly on HSCs or indirectly via action on BM niche cells.

4. The effects of antiNRP1 and antiRsp2 treatment as depicted in Figure 7e and f are very subtle, and taken at one specific time point at which the bone marrow is rapidly/exponentially recovering from the irradiation. At these timepoints (day 7 /day 10) the bone marrow cellularity is at its nadir. As depicted (i.e. without showing values for unirradiated control mice) the effects seem apparent, but if the authors would include values for unirradiated control mice it would become clear that although statistically significant, biological relevance of these very small changes are questionable.

We appreciate the Reviewer's comments and in response, we have included baseline peripheral blood WBCs and Neutrophil Counts in adult C57BL/6 mice to compare with the blood counts observed in C57BL/6 mice at day +10 after 500 cGy TBI (Fig. 7d). While we agree that the absolute magnitude of some of the differences in WBCs and Neutrophil Counts between irradiated control mice and irradiated, anti-NRP1 treated mice are small when compared to baseline WBCs and Neutrophil Counts from non-irradiated control mice, we note that the mean WBC counts in irradiated, anti-NRP1 treated mice increased approximately 83% compared to irradiated control mice (1.1 K/ μ L vs. 0.6 K/ μ L) and Neutrophil Counts increased 80% in irradiated, anti-NRP1 treated mice compared to irradiated control mice (0.54 K/ μ L vs. 0.3 K/ μ L)(Figure 7d). As importantly, Neutrophil Counts < 500 K/ μ L are strongly associated with an increased incidence of infections and increased mortality in patients (Almyroudis et al. *Transpl Infect Dis* 7, 11-17, 2005). Therefore, we respectfully submit that the accelerated recovery of Neutrophil Counts to > 500 K/ μ L by day +10 following TBI in response to anti-NRP1 treatment represents a biologically and clinically relevant finding.

5. In the discussion the authors claim that Rsp2 was essential for recovery of the HSC pool, but clearly, that conclusion cannot be drawn. HSCs recover just fine upon inhibiting of Rsp2.

We appreciate this concern and have removed the overstatement regarding the contribution of R spondin 2 to the recovery of HSCs.

6. Are there any hematopoietic data available for the mice that are shown in panel 3f? These survival curves are clear, but it would be informative to assess whether indeed peripheral blood cell recovery coincides with survival.

We appreciate this question and in response, we have measured peripheral blood complete blood count recovery in adult C57BL/6 mice at days +7, +10 and +14 following 800 cGy TBI. As shown in revised Fig. 3g, all mice in both groups displayed severely low WBCs at day +7 to day +14 following 800 cGy TBI. However, the anti-NRP1 treated mice displayed significantly increased WBCs at day +10 and increased Hgb levels at day +7 post-TBI compared to irradiated control mice. Neutrophil counts were below the level of detection for our CBC instrument in all groups of mice at these time points. In a subsequent study, we also detected increased recovery of BM KSL cells at day +10 in mice irradiated with 800 cGy TBI and treated with anti-NRP1 (Fig. 3h).

Reviewer 3

In their manuscript, Himburg and colleagues investigate the molecular mechanisms by which bone marrow (BM) endothelial cells (ECs) regulate hematopoietic regeneration upon myeloablation caused by total body irradiation (TBI) or chemotherapy. Based on their experimental data, the Authors propose a model in which, in response to myeloablation, BM ECs secrete Semaphorin 3A (SEMA3A) that inhibits BM vascular regeneration by acting via NRP1 on ECs themselves. SEMA3A/NRP1 signaling would promote EC apoptosis via Cdk5-mediated phosphorylation of p53 and increased transcription of p53-upregulated modulator of apoptosis (PUMA). Upon myeloablation, NRP1 blockade also promotes the production of Wnt signal enhancer R spondin 2 (Rspo2) in BM ECs and the expression of Rspo2 receptor LGR5 in hematopoietic stem/progenitor cells (HSPCs). Combination of Rspo2 and Wnt3a promotes hematopoietic regeneration upon TBI. This manuscript could provide interesting findings on the role of NRP1 in the control of BM hematopoiesis, however several important issues significantly hamper the strength of this work.

Major issues

1. The expression of SEMA3A in BM ECs and the ability of SEMA3A to inhibit BM angiogenesis were already reported by Vacca et al. (Blood, 2006) and Lavi et al. (Carcinogenesis, 2018), respectively. These published articles should be cited, mentioned and discussed where appropriate throughout the manuscript. Considering the existing previous work, the novelty of findings reported by Himburg et al. is lessened, yet the NRP1 link to Rspo2/LRG5 signaling is novel and previously unexplored.

We appreciate this comment and have cited both of these articles in our revised manuscript. We agree that these papers both suggest a role for SEMA3A in inhibiting angiogenesis, however we note that these findings were detected in models of multiple myeloma as opposed to normal hematopoiesis. The novelty in our principal findings is that BM ECs in healthy mice upregulate and secrete SEMA3A as an initiating lesion in response to TBI or chemotherapy, and SEMA3A, in turn, promotes BM vascular damage via binding to NRP1. These results provide new insight into a fundamental mechanism through which TBI and chemotherapy damage the BM vascular niche and delay hematopoietic recovery. We appreciate the feedback from the Reviewer on this point since it has helped us better focus the Discussion of our findings in the context of prior understanding.

2. For in vivo studies, two key reagents were employed: recombinant murine SEMA3A (R&D Systems) and goat anti-NRP1 (R&D Systems).

A. High affinity SEMA3A/NRP1 interaction depends on the binding of C-terminal SEMA3A C-end rule (C-endR) R/KXXR/K peptide(s) (Teesalu et al., PNAS 2009;), exposed by furin cleavage (Adams et al., EMBO J. 1997), to the b1 domain of NRP1 (Guo & Vander Kooi, J. Biol. Chem., 2015 and summarized in Fig. 2 of Kumanogoh & Kikutani, Nat Rev. Immunol. 2013). In the C-terminal portion of SEMA3A different furin cleavage sites exist that give rise to different furin-processed SEMA3A variants endowed with different affinity for the b1 domain of NRP1 (Guo et al., Biochemistry, 2013). A crucial aspect is that unfortunately, differently from the human version produced by the same company, the commercial recombinant mouse Sema3A-Fc Chimera from R&D Systems (Catalog Number: 5926-S3) stops at Lys 747 and lacks the last 25 amino acids that contain the two highest affinity binding sites (RAPR & RNRR; see Guo et al., Biochemistry, 2013) for the b1 domain of NRP1. Therefore, data obtained by injecting mouse Sema3A-Fc Chimera from R&D Systems are extremely difficult to be interpreted and it is difficult to reconnect the observed effects with high affinity SEMA3A binding to NRP1. To support the model they are proposing, Authors must employ a purified recombinant SEMA3A that they can prove to bind NRP1 with high affinity.

We appreciate this concern and agree to the importance of establishing the binding capacity of recombinant murine SEMA3A in light of its biochemical differences compared to the human SEMA3A. In response to this concern, we directly tested the binding affinity of the recombinant murine SEMA3A protein versus the human SEMA3A protein to immobilized murine NRP1. For this purpose, we developed a sandwich NRP1 enzyme linked immunosorbent assay (ELISA) to compare the binding capacity of recombinant murine SEMA3A and human SEMA3A (both from R&D) to the N terminus of immobilized murine NRP1. The results of this analysis are shown in revised Fig. 2a. In summary, human SEMA3A did bind to NRP1 with high affinity as expected, whereas murine SEMA3A protein also bound to NRP1 within a concentration range of 5 - 100 ng/mL. The IC₅₀ for murine SEMA3A was 35.5 ng/mL. Since we administered 2 µg (2,000 ng) SEMA3A IV every other day x 10 days in adult C57BL/6 mice, which have an expected blood volume of 2 - 2.5 mL, we believe that there is a reasonable probability that the administered dose of murine SEMA3A was sufficient to achieve concentrations of murine SEMA3A in the mouse blood to allow for binding of the administered protein with NRP1 in vivo.

B. The fact that, as reported in the data sheet, affinity-purified polyclonal goat IgGs anti-mouse/rat NRP1 from R&D Systems (Catalog Number: 5926-S3) displace the binding of both human VEGF-A 165 and human SEMA3A-Fc chimera to immobilized recombinant rat NRP1-Fc chimera clearly suggests that this polyclonal goat IgGs inhibit the binding of different ligands to NRP1 b1 domain. Since the b1 domain of NRP1 is capable of binding all ligands containing C-endR peptides and not only (e.g. He et al., Nature 2015), it is formally difficult to attribute the in vivo effect of 5926-S3 goat anti-mouse/rat NRP1 IgGs to an impaired SEMA3A binding only.

We appreciate the Reviewer's concern regarding the potential for the anti-NRP1 antibody to block both the SEMA3A binding site and the VEGF-A binding site to NRP1. In response to this concern, we have taken a genetic approach, utilizing the *Nrp1*^{VEGF-} mutant mouse, which we obtained courtesy of Dr. Chenghua Gu, Harvard University. Dr. Gu's laboratory demonstrated that a point mutation in the b1 domain of NRP1 abolishes VEGF-NRP1 interactions in this mouse (Gelfand M et al., *eLIFE* 2014;3:e03720). We compared the BM EC and hematopoietic response of *Nrp1*^{VEGF-} mice and control mice to TBI with and without anti-NRP1 antibody treatment in order to determine whether the effects of anti-NRP1 treatment were dependent on modulation of VEGF – NRP1 interactions. As shown in revised Fig. 6a, anti-NRP1 treatment significantly decreased BM EC apoptosis following TBI in *Nrp1*^{VEGF-} mice as well as control mice. Importantly, anti-NRP1 treatment also accelerated HSC regeneration in both *Nrp1*^{VEGF-} mice as well as control mice over 20 weeks post-irradiation (Fig. 6b). Taken together, these data suggest that the observed effects of anti-NRP1 treatment on BM EC survival and HSC regeneration following TBI occur independently from modulation of VEGF – NRP1 interactions. We have added comment on these points to our revised Discussion.

3. In general, the phenotypes of BM ECs and blood vessels are quite poorly analyzed and must be more carefully characterized and quantified. Authors are limiting their analyses to single sections of not otherwise defined portion(s) of the femoral BM cavity, staining only VE-cadherin, which is not an ideal marker due to its concentration at cell-to-cell contacts. BM ECs of blood vessels must be stained with DAPI, anti-endomucin, anti-CD31, and α-smooth muscle actin, and then analyzed with high resolution confocal microscopy and 3D reconstruction (e.g. see Kusumbe et al., Nat. Protocols, 2015).

We appreciate the Reviewer's concerns regarding the imaging. In response, we have performed extensive new analyses of non-irradiated and irradiated mice, treated with and without anti-NRP1 antibody, following the suggested protocol described by Kusumbe et al. *Nat Protocols* 2015. In summary, we prepared and imaged 100 micron thick sections using a Leica SP8 confocal microscope equipped with HyD single molecule detectors and deconvoluted using

the lightning deconvolution system. According to the Reviewer's suggestions, images in the revised manuscript now display DAPI, VE-Cadherin and CD31 to visualize a more comprehensive panel of vascular and cellular structures. Figures display maximum intensity z-projections to show 3D reconstructed views of the BM vasculature. We agree that these images provide significantly increased optical resolution compared to the previous widefield analyses and we are confident that our new femur preparation protocol, microscopy setup, image reconstruction, and image analysis pipelines have elevated our imaging to allow more definitive conclusions to be drawn as to the effects of modulation of SEMA3A-NRP1 signaling on the BM vasculature. We hope that the new microscopic imaging shown in our revised manuscript are satisfactory to the Reviewer.

In keeping with the Reviewer's suggestion, our revised manuscript includes new microscopic imaging of the BM vasculature, including detection of SEMA3A and NRP1, in non-irradiated mice and irradiated control mice (Fig. 1g, h), analysis of BM vascular structure, vascular area and perfusion in irradiated mice treated with anti-NRP1, isotype antibody or SEMA3A (Fig. 2b, c and Supplementary Fig. 1f), and analysis of BM vascular structure in mice treated with chemotherapy with or without anti-NRP1 or isotype antibody (Fig. 4b). Please note that we were unable to perform new microscopic imaging analysis of the *Cdh5-Cre-ERT2;Nrp1^{fl/fl}* mice or the *Cdh5-Cre-ERT2;Sema3a^{fl/fl}* mice because we moved our research laboratory from UCLA to Cedars Sinai Medical Center in the interim and this required rederivation of all of our transgenic mice strains, a process that has required several months and is still ongoing.

4. Are treatments (TBI, SEMA3A, anti-NRP1) impacting on blood vessel perfusion (e.g. use fluorescently labeled lectins) and function (e.g. monitor permeability with fluorescent dextran)?

In response to this question, we performed additional perfusion analyses using a fluorescently – labeled lectin and these new results are shown in Supplementary Fig. 1f. We did not detect any effects of TBI or modulation of SEMA3A – NRP1 signaling on BM vascular perfusion via lectin staining. We performed Evans Blue Dye staining as shown in Fig. 2h, which demonstrated that 500 cGy TBI caused significantly increased permeability of the BM vasculature compared to non-irradiated mice. Anti-NRP1 treatment following TBI significantly decreased dye extravasation in the BM compared to irradiated, isotype-treated control mice (Fig. 2h). In light of these results, we did not perform additional permeability staining with fluorescent dextran.

5. How can authors count VE-cad+ BM ECs without imaging cell nuclei?

We quantified the numbers of VEcad⁺ BM ECs based on flow cytometric analysis of percentages of VE-cad⁺ BM ECs multiplied times BM cell count numbers. We did perform DAPI nuclear staining of femur cross sections, but did not show the DAPI staining in our prior images. In all of our microscopic images shown in the revised manuscript, we have included the DAPI staining for cell nuclei. We thank the Reviewer for this recommendation.

Minor issues

On page 10, second para, referring to references 57 and 58, Authors state that SEMA3A signals via Plexin A2. It now known that both in neurons (Yaron et al., Neuron, 2005) and ECs (Kigel et al., Blood, 2011) SEMA3A mainly signals via Plexin A4 and not Plexin A2, which is instead exploited by SEMA3B (Sabag et al., J. Cell Sci., 2014).

We thank the Reviewer for pointing this out. We have revised this section of the manuscript to include these more recent references to indicate that SEMA3A primarily signals through Plexin A4.

Reviewer 4

The manuscript “Semaphorin 3A controls bone marrow vascular regeneration and R spondin 2 – mediated hematopoietic reconstitution” submitted by Himburg and colleagues studies the role of bone marrow (BM) endothelial cells (EC) in vascular and hematopoietic stem cell (HSC) regeneration after myeloablation using irradiation or chemotherapy. The authors identified that BM ECs secrete semaphorin 3A (SEMA3A) in response to myeloablation. SEMA3A binds to its receptor Neuropilin (NRP1) and induces EC apoptosis via activation of cyclin dependent kinase 5 and p53, which represses vascular regeneration. The deletion or inhibition of NRP1 reduces EC apoptosis and resulted in an accelerated BM vascular regeneration as well as HSC regeneration. Using RNA sequencing the authors identified R spondin 2 as being highly secreted in irradiated ECs upon NRP1 inhibition as well as an increased expression of the R spondin 2 receptor LGR5 in HSCs. The inhibition of R spondin 2 blocked HSC regeneration in response to NRP1 inhibition confirming the link between inhibition of SEMA3A-NRP1 signaling and secretion of R spondin 2 to promote BM EC and HSC regeneration after myeloablation.

This study aims to better understand the molecular mechanism and cellular interactions in the BM niche that mediate vascular and HSC regeneration after myeloablation. This knowledge is highly relevant to develop targeted therapies to improve vascular and HSC regeneration after irradiation or chemotherapy to reduce the time of myelosuppression in the patient. This work will be relevant to many readers in the hematopoietic field.

The presented study is well designed and easy to follow as a reader. The applied methodology represents state of the art techniques. The method section provides in a large extend enough details to reproduce the work. The data seems to be robust and supports most conclusions of the authors. However, some aspects need clarification and some additional data to support the author’s findings.

Please find these points of criticism and suggestions in the following list:

1. In Figure 1a, the expression of Cdh5 was measured in VE-Cad⁺ BM EC. It is unclear to the reader why this was done as it is not mentioned in the main text. Moreover, it is not clear why the expression is so low although presumably sorted VE-Cad⁺ BM ECs were used in the qPCR? Please comment on this aspect and mention at least once that VE-Cad and Cdh5 are the same gene. Especially for readers who are unfamiliar with ECs, this information will be helpful to better understand the applied mouse models later.

We appreciate this feedback from the Reviewer. We had originally measured *Cdh5* gene expression in VE-cad⁺ BM ECs simply to show the relatively high expression of *Nrp1* in BM ECs compared to *Cdh5*. We appreciate that measurement of *Cdh5* expression in VE-cad⁺ ECs is redundant so we have removed the *Cdh5* gene expression from revised Fig. 1b. We have edited the Results section to state that *Cdh5* is the gene that encodes VE-cadherin (page 10).

2. In Figure 1b, what is the frequency of sBMECs and aBMECs? A FACS dot plot showing the gating strategy would be helpful in the extended materials. How many cells were sorted and used for qPCR analysis? Please indicate this number in the method section.

In response to this question, we have included representative flow cytometric analysis and percentages of sBMECs and aBMECs in mice pre- and post-TBI in Supplementary Fig. 1a. We sorted 1×10^4 cells for RNA extraction for qRT-PCR analysis. We have included this information in the Methods.

3. Did the authors perform any statistics on Figure 1d? In case there is no statistical significance, please still indicate the applied statistical test and that the result is not significant.

We appreciate this question. In response, we have performed statistical analysis of differences between the cell populations shown in this Figure (originally Fig. 1d, now revised Fig. 1e). *Nrp1* expression in sBMECs and aBMECs was significantly higher than that detected in LepR⁺ stromal cells and compared to whole BM cells. We performed a Holm-Sidak's multiple comparison t-test after one-way ANOVA. This has been added to the Figure Legend for revised Fig. 1e.

4. In Figure 1h, it is not clear to the reader why endomucin⁺ ECs were included in the analysis of NRP1⁺ cells. Please comment on this aspect in the main text.

We appreciate this feedback. Type H endomucin⁺ blood vessels represent a population of capillaries with angiogenic potential (Zhao Y et al. *Ann NY Acad Sci* 1474, 5-14, 2020). Since we do not address any functional role of endomucin⁺ ECs in the paper as relates to SEMA3A-NRP1 signaling, we have removed the analysis of this population from Fig. 1j and 1k.

5. Additionally, in Figure 1h, please mention in the figure legend what the black line in the histogram refers to. Moreover, as there is a shift in the intensity of NRP1 expression an additional graph showing the MFI of NRP1 would be of great value to demonstrate increased expression of the receptor.

Thank you for this recommendation to improve this figure. In revised Fig. 1i, the black horizontal lines represent the gate for the isotype control staining (at left) and the gate for NRP1⁺ cells is to the right of the isotype gate. We have clarified this in the Fig. 1i legend. We have also added the measurements of NRP1 MFI in each group in Fig. 1k.

6. In Figure 2, the authors claim that the administration of anti-NRP1 accelerates BM vascular regeneration. However, there is no data presented showing that the BM niche is destroyed after irradiation and that it is regenerated more quickly. The presented data only shows one time point (day 7 after irradiation) at which the regeneration presumably already had occurred. It is necessary to show that the BM vasculature was damaged at earlier time points. The authors could perform a time course experiment showing the regeneration of BM vessels at different time points in the different conditions (IgG, SEMA3A and anti-NRP1). This experiment would be helpful to proof that an accelerated regeneration has occurred. Otherwise one might wonder whether anti-NRP1 administration has a protective effect.

We appreciate this suggestion from the Reviewer and agree that additional time points of evaluation would help to clarify whether anti-NRP1 treatment promotes regeneration of the BM vasculature or provides protection. In response, we have repeated the microscopic analysis of the BM vasculature to include pre-irradiation, day +3 and day +7 post-500 cGy TBI. The results of these new analyses are shown in Fig. 2b – d. In summary, at day +3 following 500 cGy TBI, all mice groups, including irradiated control mice, irradiated/SEMA3A-treated mice, and irradiated/anti-NRP1-treated mice displayed damaged and dilated BM vessels consistent with radiation injury. However, at day +7 following TBI, the anti-NRP1-treated mice demonstrated resolution of the normal BM sinusoidal vasculature, while both the irradiated control mice and the irradiated/SEMA-3A-treated mice displayed persistent BM vascular damage and vessel loss (Fig. 2c). We also show that the BM vascular area, which is a measure of BM vascular damage following TBI (Chen et al. *Cell Stem Cell* 25, 768-83, 2019), is increased in all treatment groups at day +3, but returns to normal range at day +7 only in the anti-NRP1 treatment group (Fig. 2d).

Taken together, these results suggest that anti-NRP1 treatment promotes the regeneration of the BM vasculature following injury from TBI.

7. Is it possible to score or quantify the damage of VE-Cadherin+ BM vessels that are depicted in Figure 2a and other similar figures? Maybe similar as performed in Extended Figure 3b. A score or quantification would be helpful to assess the extent of damage and regeneration after irradiation and would strengthen the findings much more than just representative images of the BM vasculature.

We appreciate this suggestion and agree. We have added measurement of BM vascular area for irradiated control mice, irradiated/SEMA-3A-treated mice and irradiated/anti-NRP1-treated mice in Fig. 2d. In a recent study by Chen et al. *Cell Stem Cell* 25, 768-83, 2019, the authors showed that 9 Gy TBI caused an increase in BM vascular area as a measurement of BM vascular damage. We have emulated this analysis and show that 500 cGy TBI increases BM vascular area and anti-NRP1 treatment significantly decreases BM vascular area at day +7 in irradiated mice to be within the range of BM vascular area in non-irradiated mice (Fig. 2d).

8. In Figure 2e, the authors write in the main text and illustrate in the figure that mice were irradiated with 700 cGy prior to the BM EC permeability assay with Evans Blue Dye. Contrarily, in the method section it is written that 500 cGy were used. Please clarify this aspect. Why were 700 cGy used although in all previous experiments 500 cGy were used? How permeable are BM ECs after 500 cGy irradiation?

We apologize for this error in labeling. The Evans Blue Dye analysis was performed on mice at +24 hours following 500 cGy TBI. The results are shown in Fig. 2h.

9. Please mention in the figure legend of Figure 3b and 3d what the dotted line refers to.

The horizontal dotted line in Fig. 3b represents the mean BM cell counts in non-irradiated control mice. We have clarified this in the Figure legend.

10. Please indicate in the figure legend of Figure 4a to what the qPCR results were normalized to.

Each sample was normalized to *Gapdh* and then the expression of each gene was normalized to the vehicle sample.

11. Please mention in the figure legend of Figure 4e what the dotted line refers to.

The dotted line in Fig. 4e represents the % Caspase 3/7⁺ ECs in non-irradiated control samples.

12. In Figure 4 and 5, to further strengthen the finding that NRP1 inhibition improves HSC regeneration it would be helpful to include a CFU Assay to measure GEMM colonies and quantification of LT-HSCs. Ultimately a competitive bone marrow transplantation as done in Figure 3a would be of great value.

In revised Fig. 4h, we have included the measurements of CFU-GEMM colonies in irradiated control mice and irradiated/anti-NRP1-treated mice to show the significantly increased CFU-GEMM colony numbers in the irradiated/anti-NRP1-treated mice. We did not perform competitive repopulation assays in the 5FU chemotherapy model.

In revised Fig. 5i, we have now included the measurements of CFU-GEMMs in the *Cdh5-Cre-ERT2;Nrp1^{fl/fl}* mice and the *Nrp1^{fl/fl}* control mice following TBI. We performed a competitive repopulation assay using BM collected at day +10 from irradiated *Cdh5-Cre-ERT2;Nrp1^{fl/fl}* mice and *Nrp1^{fl/fl}* control mice and the results are shown in Supplementary Fig. 3h. We did not detect

a difference in BM cells capable of competitive long-term repopulation between these 2 groups at day +10 post-500 cGy TBI. This may be due to a difference in biological effects on HSCs with competitive repopulating capacity in response to genetic deletion of *Nrp1* in BM ECs prior to TBI versus the functional effects of anti-NRP1 blockade administered after TBI in our model shown in Figure 2.

*13. At baseline, floxed *Sema3a* and *Nrp1* mice have different CD31+ cell counts (0.2% Figure 5e versus 0.4% Extended Data Figure 3c). Assuming that the floxed animals are on the same background how is this discrepancy explained? Is there any biological mechanism that could contribute to this observation? Or is this interexperimental variation? How sensitive is then the measurement of VE-Cad+ cells?*

The *Nrp1*^{fl/fl} mice are in a C57BL/6 background, whereas the *Sema3a*^{fl/fl} mice are in the Jcl:ICR albino mixed strain, obtained from RIKEN. We ascribe the differences in baseline CD31+ EC percentages between *Cdh5-Cre-ERT2;Nrp1*^{fl/fl} mice and *Cdh5-Cre-ERT2;Sema3a*^{fl/fl} mice to the different background strains of these mice. Flow cytometric analysis for VE-cad+ or CD31+ BM ECs is a very sensitive method to detect BM ECs and we utilize standard EC isolation procedures for flow cytometric analysis, as we have described previously in Himgurg et al. *Cell Stem Cell* 23, 370-381, 2018.

14. Please indicate in the figure legend of Figure 5a to what the qPCR results were normalized to.

The gene expression results in Fig. 5a were normalized to *Gapdh* and then normalized to the *Nrp1*^{fl/fl} control group. We have added this statement to the Fig. 5a legend.

15. In Figure 6d, in addition to Caspase3/7 activation it would be helpful to also show the frequency of Annexin V+ cells to prove presence of apoptotic cells.

Activation of Caspases 3 and 7 is a critical step in cellular apoptosis (Walsh et al. *Proc Natl Acad Sci, USA* 105;12815-12819, 2008) and sensitively discriminates apoptotic cells. In our laboratory, we have found that when analyzing cells for apoptosis following very high doses of irradiation (e.g. 800 cGy as in Fig. 6d), the activated Caspase 3/7 assay produces less background signal compared to the Annexin/7AAD analysis method. For this reason, we have utilized the activated Caspase 3/7 assay for this experiment.

16. Additionally, a one-way ANOVA was used to analyze the data in Figure 6d. The authors might use instead a two-way ANOVA here as two parameters were used.

We thank the Reviewer for this suggestion. In response, we have corrected our statistical analysis for this comparison, now shown in revised Fig. 6f, by utilizing a two-way ANOVA with Holm-Sidak's correction.

17. In Extended Data Figure 5a the expression levels of different p53 induced genes are shown. Noxa is mentioned in the main text as such a gene but the data is not presented. Is there a reason for it? Could Noxa expression being added to the figure?

We thank the Reviewer for this comment and question. We attempted to measure the gene expression of *Noxa* in BM ECs from irradiated mice, but we detected no expression of *Noxa* in BM ECs from adult C57BL/6 mice pre- or post-TBI.

18. *In contrast, Puma expression is shown twice in Figure 6e and Extended Data Figure 5a. The data does not need to be shown again in the Extended Data.*

We appreciate this suggestion and have removed *Puma* expression analysis from Supplementary Fig. 6b.

19. *In Figure 7d also Lin- and LKS- cells were analyzed but the data is not mentioned in the main text. Please mention all your results in the main text.*

We apologize for this oversight. In our revision, we focused our repeat flow cytometric analysis on BM 34-KSL HSCs for LGR5 surface expression and these results are shown in Supplementary Fig. 7a - e. Our repeat analyses using different antibodies to detect LGR5 indicates that BM HSCs express only low or absent LGR5 expression which does not increase following TBI. All data are now mentioned in the main text of the manuscript.

20. *For the cell cycle analysis in Extended Data Figure 1c, a representative FACS dot blot showing the gating strategy would be helpful for the reader to better understand the data collection.*

We have added representative flow cytometric plots to Supplementary Fig. 1d to show the percentages of BM ECs in G₀, G₁ and G₂/S/M phase.

21. *Similar to comment #3, did the authors perform any statistics on Extended Data Figure 1f, 2b-e and 3b, d-f? In case there is no statistical significance, please still indicate the applied statistical test and that the result is not significant.*

We thank the Reviewer for this guidance. In response, we have performed statistical analysis of the comparative groups in each of the noted panels. In the revision, these panels are Supplementary Fig. 2c, Supplementary Fig. 3b-e, Supplementary Fig. 4b and 4d-f. We have indicated the applied statistical test in each case in the Figure legends.

22. *Please indicate in the figure legend of Extended Data Figure 5a to what the qPCR results were normalized to.*

These data are now shown in Supplementary Fig. 6b. The gene expression results are all normalized to *Gapdh* and the non-irradiated control.

We thank the Reviewers for their guidance and believe our manuscript has been substantially improved by the revisions we have made. We hope our manuscript is now acceptable for publication in *Nature Communications*.

Reviewers' comments:

Reviewer #1 (Remarks to the Author):

None

--

Reviewer #2 (Remarks to the Author):

In the revised paper, now with a different first author, the authors have removed/withdrawn their original data in which they claimed that Lgr5 was expressed by HSCs. Apparently, their original data (which were not presented in a transparent/intuitive manner in the original manuscript) were, upon further analysis, not robust. While it makes sense to remove non-robust data, the involvement of Rsp in the biological effects that the authors observe becomes now less clear. The authors speculate on a non-Lgr5-dependent role for Rsp, but what this may be remains unclear. The mechanistic interpretation of their revised data has therefore weakened.

In the discussion the authors still state that Rsp2 was essential for recovery of the HSC pool, I do not see that this statement has been removed, or toned down, as the authors claim in their rebuttal letter.

The authors do now provide peripheral hematological data for the mice that were irradiated and treated with anti-NRP1. Although, as previously shown, the survival differences between control and anti-NRP1 treated mice are very different, the hematological parameters between both groups turn out to be essentially identical. The data provided in Figure 3g are very unlikely to explain the large differences in survival.

The authors now also show BM data for KSL cells, and argue that at day 10 progenitor counts were higher in anti-NRP1 treated mice. The example that is shown in Figure 3h (where the percentage of KLS cells amounts to 0.49%) is odd and must be an outlier, as the data in the right panel of that same figure shows overall mice range from 0,01-0,07, far away from 0,49%).

--

Reviewer #3 (Remarks to the Author):

The revised version of the manuscript is significantly improved; however, a major caveat remains.

As conceivable based on the published literature, Authors now show in Fig. 2a that the commercial recombinant mouse Sema3A-Fc chimera of R&D Systems (Catalog Number: 5926-S3) employed for their in vivo studies binds to NRP1 with a dramatically reduced affinity compared to the human SEMA3A-Fc chimera. Even if Authors do not formally provide measurements of dissociation constants, this marked drop in affinity is due to the fact that the commercial recombinant mouse Sema3A-Fc chimera from R&D Systems stops at Lys 747 and lacks the last 25 amino acids that contain the two highest affinity binding sites (RAPR & RNRR; see Guo et al., *Biochemistry*, 2013, 52:7551-7558) for the b1 domain of NRP1.

If Authors want to propose a model according to which the in vivo effects observed upon Sema3A treatment involve NRP1, they must inject in animals a Sema3A recombinant protein that binds with physiological (i.e. high) affinity to NRP1 in vitro. This is not the case, at present. Authors need to perform all in vivo experiments with a Sema3A capable of binding NRP1 with physiological affinity, i.e. a Kd for Sema3A binding to NRP1 around 1.5 nM (see He & Tessier-Lavigne, *Cell*, 1997, 90:739-751; Kolodkin et al., *Cell*, 1997, 90:753-762). The rebuttal statement "we believe that there is a reasonable probability that the administered dose of murine SEMA3A was sufficient to achieve

concentrations of murine SEMA3A in the mouse blood to allow for binding of the administered protein with NRP1 in vivo" is a scientifically weak argument.

--

Reviewer #4 (Remarks to the Author):

The authors addressed most questions and suggestions raised by the reviewers and added valuable data and information to support and strengthen their findings.

Response to Reviewers

Reviewer 1

No comments

Reviewer 2

In the revised paper, now with a different first author, the authors have removed/withdrawn their original data in which they claimed that Lgr5 was expressed by HSCs. Apparently, their original data (which were not presented in a transparent/intuitive manner in the original manuscript) were, upon further analysis, not robust. While it makes sense to remove non-robust data, the involvement of Rsp in the biological effects that the authors observe becomes now less clear. The authors speculate on a non-Lgr5-dependent role for Rsp, but what this may be remains unclear. The mechanistic interpretation of their revised data has therefore weakened.

The change in first author was a function of the departure of Dr. Heather Himburg, prior first author, from UCLA to the Medical College of Wisconsin to begin her independent faculty career, which occurred during the interim period when we were revising the manuscript in response to the prior review. Since Dr. Himburg was not able to complete the additional experiments and analyses that were necessary for our revision of this manuscript, she graciously offered Dr. Christina Termini, post-doctoral fellow in my laboratory, the opportunity to complete the important revision work and take on the role as first author. Dr. Himburg remains an author on our revised manuscript, as the penultimate author.

We respectfully disagree with the concern that the demonstration of the role of R spondin 2 in mediating hematopoietic regeneration is “less clear”. In Figure 7b and 7c, we demonstrated that BM ECs in irradiated mice upregulated expression and secretion of R spondin 2 in response to anti-NRP1 treatment. We further demonstrated that systemic administration of a blocking anti-R spondin 2 antibody suppressed anti-NRP1-mediated recovery of peripheral blood WBCs, Neutrophils, BM CD150⁺CD48⁻KSL HSCs and BM KSL HSPCs (Figure 7d – 7f). In keeping with these findings, administration of anti - R spondin 2 suppressed BM colony forming cell (CFC) recovery and CFU-GEMM recovery in irradiated mice treated with anti-NRP1 (Figure 7g). Finally, we also demonstrated that R spondin 2 – mediated recovery of CFU-GEMMs from irradiated BM KSL cells was dependent on co-activation with Wnt3a (Figure 7h). Since R spondins have not been previously described to have a role in regulating hematopoietic regeneration, these results add significantly to the field and provide the foundation for new studies into the function of R spondins in regulating hematopoietic regeneration.

We appreciate the Reviewer’s comment that we have not yet determined the precise receptor or receptors through which R spondin 2 mediates hematopoietic regeneration and agree that this aspect of mechanistic insight will be important to unravel in the future. However, since R spondins are well understood to have the capacity to signal through multiple receptors, including leucine-rich repeat containing G protein-coupled receptors (LGRs), RNF43 and ZNRF3 transmembrane ligases, Syndecan 4, low-density lipoprotein receptor-related protein 6 (LRP6), and frizzled 8 (FZD8), and to mediate cellular effects through heparan sulfate proteoglycan – dependent (HSPG) signaling in the absence of LGRs (de Lau W, et al. *Genome Biology* 2012;13:242; Ohkawara B, et al. *Dev Cell* 2011;20:303-14; Dubey R et al. *eLife* 2020;9:e54469; Szenker-Ravi E, et al. *Nature* 2018;557:564-571; Kim K, et al. *Mol Biol Cell* 2008;19:2588-96;

Nam J, et al. *J Biol Chem* 2006;281:13247-57), the determination of the receptor(s) through which R spondin 2 promotes hematopoietic regeneration will require multipronged investigations which we believe are beyond the scope of this manuscript. It is also possible that R spondin 2 may be promoting hematopoietic regeneration via indirect actions on other BM niche cells that also contribute to the hematopoietic regenerative process, which will also require additional cell-specific genetic studies in mice.

We want to emphasize again that the major mechanistic insights in this manuscript are: 1) our discovery of the crucial role of SEMA3A – NRP1 signaling in BM ECs in regulating the BM vascular response to myelosuppression, 2) the demonstration that genetic or pharmacologic inhibition of SEMA3A – NRP1 signaling in ECs suppresses p53-mediated BM EC apoptosis and accelerates BM vascular regeneration, 3) the demonstration that NRP1 regulation of BM vascular regeneration occurs independently from VEGF-NRP1 signaling, and 4) the discovery that inhibition of SEMA3A – NRP1 signaling in ECs accelerates HSC regeneration and hematopoietic reconstitution in mice following myelosuppression. These results add importantly to the fundamental understanding of the mechanisms that control BM vascular niche regeneration following myelosuppression, which is essential for HSC regeneration and hematopoietic reconstitution to occur. Furthermore, we have demonstrated that the SEMA3A-NRP1 signaling pathway can be therapeutically targeted to accelerate vascular regeneration and hematopoietic reconstitution in vivo. We believe these central mechanistic findings will have significant impact and merit publication as stand-alone findings.

In the discussion the authors still state that Rsp2 was essential for recovery of the HSC pool, I do not see that this statement has been removed, or toned down, as the authors claim in their rebuttal letter.

We thank the Reviewer for bringing this to our attention. This was an unintentional error and we have removed this statement from the Discussion.

The authors do now provide peripheral hematological data for the mice that were irradiated and treated with anti-NRP1. Although, as previously shown, the survival differences between control and anti-NRP1 treated mice are very different, the hematological parameters between both groups turn out to be essentially identical. The data provided in Figure 3g are very unlikely to explain the large differences in survival.

We thank the Reviewer for this comment and appreciate that the magnitudes of differences in peripheral blood WBC and hemoglobin levels between 800 cGy-irradiated control mice and 800 cGy-irradiated, anti-NRP1 treated mice, shown in Figure 3g, are relatively modest compared to the substantial differences in animal survival shown in Figure 3f. Analyses of peripheral blood counts in mice over days +7 to +14 following 800 cGy TBI can be problematic because numerous mice in the irradiated, control group predictably die rapidly from day +10 onward and we are not able to capture peripheral blood counts on all of these mice since the mice are not observed continuously during this period. As such, the sensitivity of detecting differences in peripheral blood counts between day +10 to day +14 in 800 cGy-irradiated mice is much lower than when sublethal irradiation doses (e.g. 500 cGy TBI as shown in Figure 3a) are employed. Nonetheless, the results shown in Figure 3h, which demonstrate that 800 cGy-irradiated, anti-NRP1 treated mice displayed a doubling in percentages of BM c-Kit⁺Sca-1⁺Lin⁻ (KSL) HSPCs compared to 800 cGy-irradiated control mice does contribute to an explanation for the substantial differences in survival of these mice groups following 800 cGy irradiation. Yang et

al. (*Blood* 2005;105:2717-2723) previously showed that the BM cells most enriched for radioprotective cells capable of rescuing mice from lethal irradiation were the KSLCD34⁺flt3⁻ short-term HSCs. Since we have demonstrated a doubling in the percentages of BM KSL cells at day +10 following 800 cGy TBI in response to anti-NRP1 treatment, and since BM KSL cells contain a substantial percentage of the most highly radioprotective BM KSLCD34⁺flt3⁻ cells, these data support the conclusion that anti-NRP1 treatment promotes the recovery of BM radioprotective cells that are essential for mice survival following lethal irradiation.

The authors now also show BM data for KSL cells, and argue that at day 10 progenitor counts were higher in anti-NRP1 treated mice. The example that is shown in Figure 3h (where the percentage of KLS cells amounts to 0.49%) is odd and must be an outlier, as the data in the right panel of that same figure shows overall mice range from 0,01-0,07, far away from 0,49%.

We thank the Reviewer for pointing out this typographical error in our labeling of the representative flow cytometric plot shown in the prior version of Figure 3h. The number originally shown to represent the percentage of cells in the KSL gate, 0.49%, was a simple error and should have been 0.049%. We have corrected this labeling error in the revised Figure 3h panel. The numbers shown in the representative flow cytometric plot are consistent with and represented on the bar graph on the right in revised Figure 3h.

Reviewer 3

The revised version of the manuscript is significantly improved; however, a major caveat remains.

*As conceivable based on the published literature, Authors now show in Fig. 2a that the commercial recombinant mouse Sema3A-Fc chimera of R&D Systems (Catalog Number: 5926-S3) employed for their in vivo studies binds to NRP1 with a dramatically reduced affinity compared to the human SEMA3A-Fc chimera. Even if Authors do not formally provide measurements of dissociation constants, this marked drop in affinity is due to the fact that the commercial recombinant mouse Sema3A-Fc chimera from R&D Systems stops at Lys 747 and lacks the last 25 amino acids that contain the two highest affinity binding sites (RAPR & RNRR; see Guo et al., *Biochemistry*, 2013, 52:7551-7558) for the b1 domain of NRP1.*

*If Authors want to propose a model according to which the in vivo effects observed upon Sema3A treatment involve NRP1, they must inject in animals a Sema3A recombinant protein that binds with physiological (i.e. high) affinity to NRP1 in vitro. This is not the case, at present. Authors need to perform all in vivo experiments with a Sema3A capable of binding NRP1 with physiological affinity, i.e. a Kd for Sema3A binding to NRP1 around 1.5 nM (see He & Tessier-Lavigne, *Cell*, 1997, 90:739-751; Kolodkin et al., *Cell*, 1997, 90:753-762). The rebuttal statement “we believe that there is a reasonable probability that the administered dose of murine SEMA3A was sufficient to achieve concentrations of murine SEMA3A in the mouse blood to allow for binding of the administered protein with NRP1 in vivo” is a scientifically weak argument.*

We thank the Reviewer for the positive feedback that the manuscript is significantly improved.

We appreciate the concern about utilizing a recombinant SEMA3A with high affinity binding to NRP1 for the in vivo studies. In response, we have pursued several avenues to obtain a recombinant SEMA3A protein which we can demonstrate to have high affinity binding to NRP1.

First, we requested custom production of the full length murine SEMA3A recombinant protein by R&D Systems (Biotechne) since R & D Systems manufactures the full length human SEMA3A Fc protein and the truncated murine SEMA3A Fc protein for commercial use. R&D Systems informed us that R&D Systems scientists had previously attempted to produce the full length murine SEMA3A protein and were unsuccessful due to degradation of C terminus residues by furin-type proteases. R&D Systems scientists further advised that production of a full length murine SEMA3A protein would not be feasible without performing site mutagenesis experiments that would require 9-12 months in their experienced hands. We subsequently consulted other custom protein production companies that provided the same feedback.

We next pursued the production of a rat SEMA3A protein via custom production by CUSABIO (Houston, TX). After procurement of the recombinant rodent SEMA3A protein from CUSABIO, we tested this product for affinity binding to murine NRP1 using surface plasmon resonance (SPR) which provides a high sensitivity technique to detect molecular binding (Alvarado C, et al. *Molecules* 24(18):3552, 2019; Capelli D, et al. *Front Chem* 7:1-14, 2020). This analysis was performed via collaboration with Dr. Ramachandran Murali, Associate Professor of Biochemistry, Cedars Sinai Medical Center. Briefly, using a Pioneer SensiQ SPR instrument, rat SEMA3A Fc was resuspended in 20 mM HEPES pH 5.0, 150 mM NaCl and 0.005% Tween-20 and murine NRP1 was immobilized on a PCH biosensor chip in 10 mM sodium acetate pH 5.0 to desired RU. Rat SEMA3A Fc protein was flowed over the immobilized mouse NRP1 to measure binding affinities. OneStep kinetics was used for affinity measurements, wherein the analyte was injected at a single fixed concentration (50 or 100 nM Rat SEMA3A at 75 μ l/min flowrate). Binding analysis and affinity calculations were performed using Pioneer QDAT software. This analysis demonstrated that the rat SEMA3A Fc protein failed to bind to murine NRP1 (**Figure 1 below**).

Figure 1. SPR binding assay for rat SEMA3A Fc with murine NRP1

In light of the SPR binding assay indicating absent binding capacity for the rat SEMA3A Fc protein with murine NRP1, and with no feasible option to produce or obtain a full length murine SEMA3A Fc protein, we focused on characterizing the full length human SEMA3A Fc protein, produced by R&D Systems, which has high amino acid sequence homology to full length murine SEMA3A, including the high affinity binding residues in the C terminal region, and which we previously demonstrated to have high affinity binding to murine NRP1 in vitro via a colorimetric labeling approach (**Figure 2 below**).

Figure 2. Affinity binding assay for full length human SEMA3A Fc (blue) with murine NRP1 (in vitro labeling approach). Binding of truncated murine SEMA3A Fc (green) to murine NRP1 is also shown.

We next procured larger amounts of the human SEMA3A Fc protein from R & D Systems and tested the affinity binding of this protein for immobilized murine NRP1 via the SPR assay. Full length human SEMA3A Fc bound to murine NRP1 with high affinity ($K_D=1.8$ nM), in the physiologic range noted by the Reviewer. We also measured the binding affinity of the truncated murine SEMA3A Fc protein (R & D Systems) by SPR and this assay demonstrated lower binding to murine NRP1, with a $K_D=4.4$ nM (**Figure 3 below**). These results are included in the revised manuscript, **Supplementary Fig. 2c and 2d**.

Figure 3. SPR binding assay for full length human SEMA3A Fc with murine NRP1. SPR assay for truncated murine SEMA3A Fc protein binding with murine NRP1 is also shown.

Since the full length human SEMA3A Fc protein has high sequence homology to full length murine SEMA3A, contains the high affinity binding residues critical for NRP1 binding, RNRR and RAPR (see **Supplementary Fig. 2a**), and displays the requisite high binding affinity to murine NRP1 in the SPR assay, we repeated our in vivo studies of SEMA3A effects on the BM vasculature and hematopoietic recovery in irradiated mice using the human SEMA3A Fc protein. As shown in **Supplementary Fig. 2e**, treatment of irradiated mice with 2 μ g human SEMA3A Fc intraperitoneally every other day from day +1 to day +10 did not significantly alter the BM vasculature or BM vascular area compared to irradiated control mice. Administration of human SEMA3A FC to irradiated mice also did not affect peripheral blood WBC, Neutrophil or Lymphocyte Counts compared to irradiated control mice, and had no significant effects on BM cell counts, percentages of BM c-Kit⁺Sca-1⁺lin⁻ (KSL) hematopoietic stem/progenitor cells

(HSPCs) or numbers of BM KSL cells (**Supplementary Fig. 4a-d**). As we state in the manuscript text, we hypothesize that systemic administration of recombinant SEMA3A to irradiated mice did not worsen BM vascular damage or hematopoietic toxicities because high levels of endogenous SEMA3A secreted by BM ECs in irradiated mice (see **Fig. 1f**) likely obscured potential effects of exogenously administered SEMA3A.

Since our central hypothesis was that inhibition or blockade of SEMA3A – NRP1 signaling in BM ECs would promote BM vascular regeneration and, in turn, hematopoietic regeneration, we have revised main **Figure 2** and **Figure 3** to focus on the principal comparison of irradiated, anti-NRP1 treated mice versus irradiated, isotype-treated control mice. Our results utilizing the blocking anti-NRP1 antibody have been validated by the EC-specific *Nrp1*- and *Sema3a*-knockout studies in irradiated mice, each of which comparably accelerated BM vascular recovery and hematopoietic regeneration in mice (**Figures 2, 3, 5** and **Supplementary Fig. 7**).

We hope that the new data demonstrating the high affinity binding of the human SEMA3A Fc protein to murine NRP1 and the effects of administration of this high affinity binding SEMA3A in irradiated mice will be satisfactory to the Reviewer. This was the only feasible option available to us to test a high affinity binding SEMA3A in our mouse model.

Reviewer 4

The authors addressed most questions and suggestions raised by the reviewers and added valuable data and information to support and strengthen their findings.

We thank the Reviewer for these positive comments on our manuscript.

REVIEWERS' COMMENTS

Reviewer #2 (Remarks to the Author):

The authors have addressed most of my concerns

Reviewer #3 (Remarks to the Author):

The authors have satisfactorily addressed all my remaining concerns.

Point – by – Point Response to Reviewers

Reviewer 2

The authors have addressed most of my concerns.

Response: We thank the Reviewer for this feedback on our revised manuscript.

Reviewer 3

The authors have satisfactorily addressed all my remaining concerns.

Response: We thank the Reviewer for this feedback.